# Centrosome amplification and aneuploidy driven by the HIV-1-induced Vpr•VprBP•Plk4 complex in CD4+ T cells

Jung-Eun Park [1,10], Tae-Sung Kim [1,10], Yan Zeng[1], Melissa Mikolaj[2,3], Jong Il Ahn [1], Muhammad S. Alam[4], Christina M. Monnie[5], Victoria Shi[6], Ming Zhou [7], Tae-Wook Chun[6], Frank Maldarelli [8], Kedar Narayan [2,3], Jinwoo Ahn [5], Jonathan D. Ashwell [4], Klaus Strebel[9] & Kyung S. Lee [1] ✉

HIV-1 infection elevates the risk of developing various cancers, including T-cell lymphoma. Whether HIV-1-encoded proteins directly contribute to oncogenesis remains unknown. We observe that approximately 1–5% of CD4+ T cells from the blood of people living with HIV-1 exhibit over-duplicated centrioles, suggesting that centrosome amplification underlies the development of HIV-1-associated cancers by driving aneuploidy. Through affinity purification, biochemical, and cellular analyses, we discover that Vpr, an accessory protein of HIV-1, hijacks the centriole duplication machinery and induces centrosome amplification and aneuploidy. Mechanistically, Vpr forms a cooperative ternary complex with an E3 ligase subunit, VprBP, and polo-like kinase 4 (Plk4). Unexpectedly, however, the complex enhances Plk4's functionality by promoting its relocalization to the procentriole assembly and induces centrosome amplification. Loss of either Vpr's C-terminal 17 residues or VprBP acidic region, the two elements required for binding to Plk4 cryptic polo-box, abrogates Vpr's capacity to induce these events. Furthermore, HIV-1 WT, but not its Vpr mutant, induces multiple centrosomes and aneuploidy in human primary CD4+ T cells. We propose that the Vpr•VprBP•Plk4 complex serves as a molecular link that connects HIV-1 infection to oncogenesis and that inhibiting the Vpr C-terminal motif may reduce the occurrence of HIV-1-associated cancers.

A large body of evidence suggests that people living with HIV-1 are at high risk of developing various comorbid diseases, including cancer[1]. While the weakened immune system brought about by HIV-1 infection is generally blamed for the increased risk of developing these disorders, several studies suggest that integration of HIV-1 proviruses in oncogenes could promote cellular transformation and the development of T-cell lymphomas[2–5]. Interestingly, centrosome amplification, which is prevalent among hematological malignancies[6–9], constitutes the major causal mechanism of chromosomal instability and aneuploidy[10]. These observations raise the possibility that HIV-1-encoded proteins may promote oncogenesis by deregulating the centrosome duplication process.

As the main microtubule-organizing center in animal cells, the centrosome (composed of a pair of barrel-shaped centrioles and their surrounding pericentriolar material) is critically required for normal cell division and proliferation[11,12]. Tight control of centrosome number is fundamentally required for proper bipolar spindle formation, an event critical for accurate chromosome segregation and the maintenance of genomic integrity. Studies with cultured cells show that HIV-1 Vpr (viral protein R), a multifunctional molecular adapter[13–15],

localizes to the centrosome and can induce centrosome over-duplication (i.e., more than two centrosomes per cell)[16,17]. The mechanism underlying how Vpr induces centrosome overduplication at the molecular level remains unknown. Notably, Vpr binds to various cellular proteins[13], including VprBP/RIP, a high-affinity target of HIV-1 Vpr[18] with promiscuous scaffold functions[19]. (VprBP is also named DDB1-CUL4-Associated Factor 1 [DCAF], a substrate receptor for the CUL4-DDB1 ubiquitin ligase[20,21]). Several studies demonstrate that VprBP functions as a substrate recognition subunit of E3 ubiquitin ligases[19,22,23] and that VprBP interaction with Vpr or the structurally related Vpx is vital to detect specific cellular targets for their protea-somal degradation[24–27]. Recent studies show that, in addition to loca-lizing to the nucleus, VprBP localizes to centrosomes and promotes the degradation of CP110[17,28], a conserved centriolar protein required for centrosome duplication[29,30]. Whether VprBP has an uncharacterized role other than serving as a component of E3 ubiquitin ligases has yet to be discovered.

Polo-like kinase 4 (Plk4) is a key regulator of centriole biogenesis[31–33], which occurs precisely once per cell cycle[10]. Plk4 is recruited to the outmost region of a pericentriolar scaffold Cep152 in early G1[34–37] through the interaction between its non-catalytic cryptic polo-box (CPB) domain and the N-terminal region of Cep152[37]. As the level of Plk4 rises in late G1, it undergoes *trans*-autophosphorylation-dependent[38] liquid–liquid phase separation (LLPS)[39–42] and dynamic relocalization from around Cep152 (i.e., ring-like state) to the procen-triole assembly site (i.e., dot-like state)[36,43]. Dysregulation of this pro-cess results in abnormal centrosome numbers and chromosome instability that could lead to aneuploidy, a cause of cancer development[10,44–46].

Here we show that the CD4[+] T cells purified from the blood of people living with HIV-1, but not healthy individuals, exhibit over-duplicated centrosomes in approximately 1–5% of the population. Subsequent analyses suggest that centrosome amplification is driven by the ability of Vpr to form a cooperative complex with VprBP and Plk4 and induce Plk4-mediated centriole overduplication. At the molecular level, Vpr, which binds to the VprBP WD40 domain[27,47], interacts with Plk4 CPB through its C-terminal tail (CT; residues 80–96). The C-terminal acidic region (AR) of VprBP also interacts with the CPB of Plk4, establishing three-way interactions to form the Vpr•VprBP•Plk4 complex. Consistent with these findings, deletion of either the Vpr CT or VprBP AR significantly diminishes the level of Vpr-induced centrosome overduplication and aneuploidy in various CD4[+] cells, including primary T cells prepared from peripheral blood mononuclear cells (PBMCs) of healthy human subjects. Given that a modest elevation of Plk4 level (<2-fold) is sufficient to induce various tumors in a mouse model[48], the data provided here suggest that the HIV-1-induced Vpr•VprBP•Plk4 complex can promote oncogenesis by bolstering Plk4-dependent centriole duplication.

## Results
### Centrosome amplification in CD4[+] T cells from people living with HIV-1
To explore whether HIV-1 can induce centrosome abnormalities, we performed immunostaining analyses with primary CD4[+] T cells pur-ified from the PBMCs of 14 healthy individuals and 10 individuals living with HIV-1 (before they developed HIV-1-associated disorders, includ-ing cancer) (Supplementary Fig. 1a–e). Doubly stained Cep152 and γ-tubulin signals were used as surrogate markers for a centrosome (Supplementary Fig. 1a). Control CD4[+] T cells purified from 14 healthy individuals showed less than 0.05% of overduplicated centrosomes ($n = 8668$) (Supplementary Fig. 1b, c). To our surprise, although the degree of centrosome overduplication varied from 0.9% to 5.1% among different samples, the occurrence of cells with overduplicated cen-trosomes was manifest in all 10 cases examined (Fig. 1a, b and Sup-plementary Fig. 1d, e). Remarkably, all four samples obtained after

0.4–2.8 years of antiretroviral therapy (ART) exhibited a significantly diminished level of overduplicated centrosomes when compared to those before the ART (Fig. 1c and Supplementary Fig. 1d). These find-ings suggest that HIV-1 infection may alter cellular processes that can lead to centrosome amplification and aneuploidy, a condition that can drive oncogenesis[45,46]. Therefore, since Plk4 is a master regulator of centrosome duplication, we investigated whether its function is deregulated under HIV-1 infection.

### VprBP AR interacts with Plk4 CPB
To identify novel Plk4-binding proteins whose function could be influenced by HIV-1, we performed two independent affinity purification–mass spectrometry analyses using HEK293T cells expressing Plk4 CPB (581–884) or the entire C-terminal domain (CTD) (581–970). The results showed that both CPB and CTD effectively precipitated VprBP, a major HIV-1 Vpr-binding protein[18], and its asso-ciating DDB1, a subunit of VprBP-mediated E3 ligase complexes[25,27,28,49,50] (Fig. 2a). As expected, Plk4, which forms a homodimer[37,51], and other proteins known to interact/associate with Plk4 (such as Cep152, Cep192, PCNT, Cep63, Cep135, and Cep57)[34–36,52–55] were also copurified (Supplementary Fig. 2a). Under these conditions, CUL4A and CUL4B, the subunits of E3 ubiquitin ligase complexes[13,19,22,23], were not significantly detected (Fig. 2a). Consistently, coimmunoprecipitation analyses carried out with trans-fected HEK293T cells showed that the full-length Plk4 efficiently coprecipitated VprBP under thymidine-treated (S-phase) or nocodazole-treated (G2/M-phase) conditions (Supplementary Fig. 2b). It also interacted with DDB1, albeit at a reduced level (Supplemen-tary Fig. 2c).

In accord with the data in Fig. 2a, both CTD and CPB of Plk4, but not the N-terminal domain (NTD), effectively interacted with the full-length VprBP (Fig. 2b). In a reverse experiment, the AR (residues 1401–1507) of VprBP coprecipitated the full-length Plk4, whereas its partially deleted AR fragments (i.e., 1401–1470 and 1401–1450) exhib-ited a significantly compromised or an undetectable level, respec-tively, of Plk4 binding (Fig. 2c). VprBP(1446–1507) appeared to be sufficient for the VprBP–Plk4 interaction (Fig. 2d). Consistent with these observations, a recombinant His-Maltose binding protein (MBP)-fused VprBP(1446–1507) expressed in *E. coli* efficiently interacted with CPB (Supplementary Fig. 2d), and the two proteins coeluted from size-exclusion chromatography (SEC) (Fig. 2e). Taken together, these data suggest that VprBP forms a binary complex with Plk4 through the interaction between its AR and Plk4 CPB.

### HIV-1 Vpr, but not HIV-2 Vpr or Vpx, forms a cooperative complex with VprBP and Plk4
As one of the main cellular binding targets for HIV-1 Vpr, VprBP is shown to bind to Vpr through the interactions between its WD40 domain and Vpr's N-terminal and α3 region[27]. As expected, Vpr inter-acted with the WD40-containing fragments but failed to interact with the AR(1401–1507) fragment (Fig. 3a) (all Vpr constructs were derived from the HIV-1 NL4-3 isolate, unless indicated otherwise). To deter-mine whether Vpr influences the VprBP–Plk4 interaction, we per-formed coimmunoprecipitation analyses using two VprBP constructs containing the Vpr-binding WD40 domain. Remarkably, the coex-pression of Vpr enhanced the VprBP–Plk4 interaction (Fig. 3b and Supplementary Fig. 3a) in an expression-level-dependent manner (Fig. 3c and Supplementary Fig. 3b). This suggests that Vpr can induce the formation of a ternary complex with VprBP and Plk4. The catalytic activity of Plk4 was not required for Vpr-enhanced VprBP–Plk4 inter-action (Supplementary Fig. 3c). In an experiment carried out with STREP-Vpr as a ligand, the ability of VprBP to augment the Vpr–Plk4 interaction was manifest (Supplementary Fig. 3d). In addition, the Vpr(1–82) truncate lacking the CT region exhibited a greatly impaired ability to interact with Plk4, although it largely maintained the level of

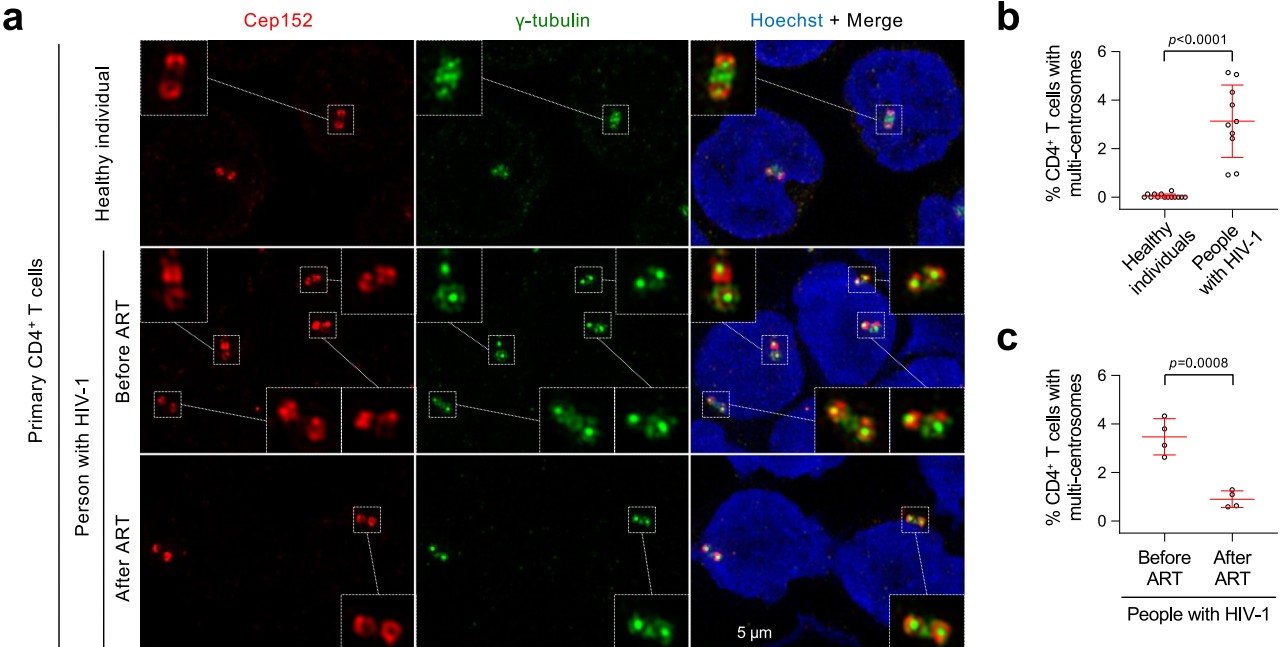

**Fig. 1 | Centrosome amplification in the primary CD4⁺ T cells purified from people living with HIV-1. a** Representative three-dimensional structured illumination microscopy (3D-SIM) images showing over-duplicated centrosomes (marked by Cep152 and γ-tubulin signals) in primary CD4⁺ T cells purified from healthy individuals and people living with HIV-1. Purified cells were immunostained immediately without culturing them. Boxes, area of enlargement. **b** Data showing the percentage of CD4⁺ T cells that exhibit multiple centrosomes. Quantification was performed with the cells purified from the blood of 14 healthy people [a total of $n = 14$ biologically independent samples examined over two independent experiments (1st set: samples 1–2 and 2nd set: samples 3–14 in Supplementary Fig. 1c)] and 10 people living with HIV-1 [a total of $n = 10$ biologically independent samples examined over two independent experiments (1st set: samples 1–5 and 2nd set: samples 6–10 in Supplementary Fig. 1d, e)]. **c** Quantified data showing the reduction of primary CD4⁺ T cells with multiple centrosomes after ART. The data were obtained by analyzing four paired samples (a total of $n = 8$ biologically independent samples) obtained before and after ART from the same people with HIV-1 (the #1–4 individuals listed in Supplementary Fig. 1d) Bars, mean ± s.d.; $P$ values, unpaired two-tailed $t$-tests. Detailed clinical data for all the samples analyzed here are provided in Supplementary Fig. 1c–e.

VprBP binding (Fig. 3d). The C-terminal Vpr(51–96) and Vpr(75–96) fragments exhibited a capacity to interact with Plk4 at a low level (Fig. 3d). Unlike Vpr (Lai, NL4-3, and 89.6), several variants of structurally related Vpx[27] and HIV-2 Vpr exhibited either a low or undetectable level of binding to VprBP and failed to enhance the VprBP–Plk4 interaction (Supplementary Fig. 3e–g). In addition, although mitotic Plk1 weakly (approximately threefold weaker than Plk4) interacted with VprBP, Vpr failed to augment this interaction (Supplementary Fig. 3h). Further analysis with a recombinant Vpr showed that it directly interacted with CPB or CTD in vitro (Supplementary Fig. 3i). These findings and the data in Fig. 2 suggest that Vpr, VprBP (WD40-AR), and Plk4 CPB directly interact with one another and generate a cooperative ternary complex.

To investigate the nature of the ternary complex, SEC was carried out with purified proteins (i.e., Vpr, VprBP WD40-AR, Plk4 CPB, and DDB1). Analysis of fractionated samples suggests the formation of a ternary Vpr•VprBP•Plk4 complex with an approximately 1:1:2 stoichiometry (Fig. 3e). (Plk4 functions as a homodimer[37,51].) DDB1, which is shown to bind to VprBP[27], also copurified with the ternary complex (Fig. 3e). Further analyses with interferometric scattering mass spectrometry (iSCAMS), which detects protein–protein interactions in a real-time, revealed that, at 200 mM NaCl, the VprBP (WD40-AR)•Vpr complex (Fig. 3f, 1st panel, red arrow) generated a ternary complex with Plk4 CPB (Fig. 3f, 3rd panel, thick black arrow) immediately after mixing (in less than a minute). At 400 mM NaCl, however, the ternary complex became largely dissociated (Fig. 3f, 4th panel), presumably because the AR region, which is heavily enriched in Asp and Glu residues with the calculated pI of 2.4, cannot stably interact with the basic CPB (pI of 9.1) in a high-ionic-strength environment.

## Plk4 is not the target of the Vpr•VprBP-mediated E3 ubiquitin ligase

Various studies have shown that Vpr forms a complex with a cullin-4-RING E3 ubiquitin ligase (i.e., the CRL4-VprBP-Vpr complex) and regulates multiple intracellular target proteins through proteasomal degradation[13,19,22,23]. However, overexpression of Vpr, VprBP, or both did not detectably alter Plk4 stability, whereas it effectively induced degradation of a previously characterized substrate, Uracil DNA Glycosylase-2 (UNG2)[49] (Supplementary Fig. 3j,k). In addition, while depletion of β-TrCP, a known F-box protein for Plk4[56], increased the steady-state level of Plk4, depletion of VprBP failed to noticeably change the level of Plk4 (Supplementary Fig. 3l). In in vitro ubiquitination assays carried out with purified proteins, Vpr did not appear to influence the level of ubiquitinated Plk4 (Supplementary Fig. 3m, top). Under the same conditions, however, ubiquitination of UNG2 was enhanced even by the Vpr(1–79) form lacking the CT (Supplementary Fig. 3m, bottom), as reported previously[49]. These observations suggest that the binding of Vpr and VprBP may not prompt proteasomal degradation of Plk4.

## Vpr and VprBP colocalize with Plk4 and promote Plk4's ring-to-dot conversion around a centriole

Consistent with the previous findings[28,57], VprBP localized to the nucleus and centrosomes (Supplementary Fig. 4a) with its signals found at or near a region where Cep152 localizes (Fig. 4a). Since VprBP AR interacted with Plk4 CPB (Fig. 2), we investigated whether VprBP influences Plk4's pericentriolar localization in an AR-dependent manner. In U2OS cells, depletion of endogenous VprBP by RNAi mildly reduced the number of the dot-state Plk4, and the expression of RNAi-insensitive VprBP, but not the VprBPΔAR mutant (residues 1–1427),

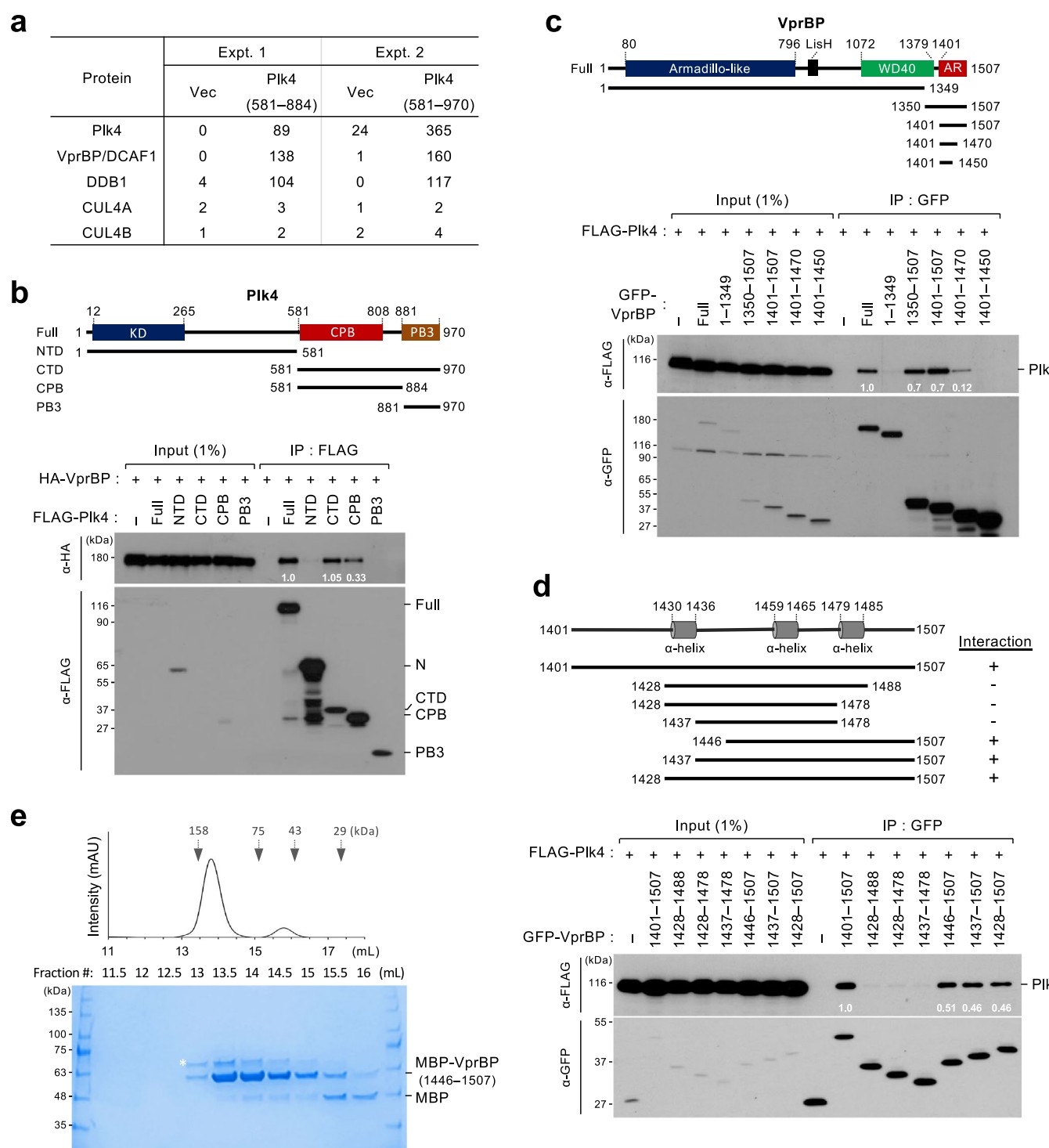

**Fig. 2 | Direct interaction between VprBP AR and Plk4 CPB. a** Two independent mass spectrometry analyses showing that VprBP and DDB1 co-purified with the indicated Plk4 ligands. The peptide counts of the listed proteins co-purified with the ligands are shown. A ZZ-tagged Plk4 CPB (581–884) and a FLAG-tagged Plk4 CTD (581–970) were used as ligands for Expt. 1 and 2, respectively (see Methods for details). **b–d** Coimmunoprecipitation analyses performed with HEK293T cells transfected with the indicated constructs. Residue numbers are shown in the schematic diagram. KD kinase domain, CPB cryptic polo-box domain, PB3 polo-box 3: NTD, N-terminal domain, CTD C-terminal domain, WD40 WD40 domain, AR Acidic region. Numbers in the immunoblots represent relative signal intensities. **e** SEC profile and SDS-PAGE showing coeluting MBP-VprBP(1446–1507) and CPB(581–808) in the Coomassie Brilliant Blue-stained gel. Asterisk, contaminating protein.

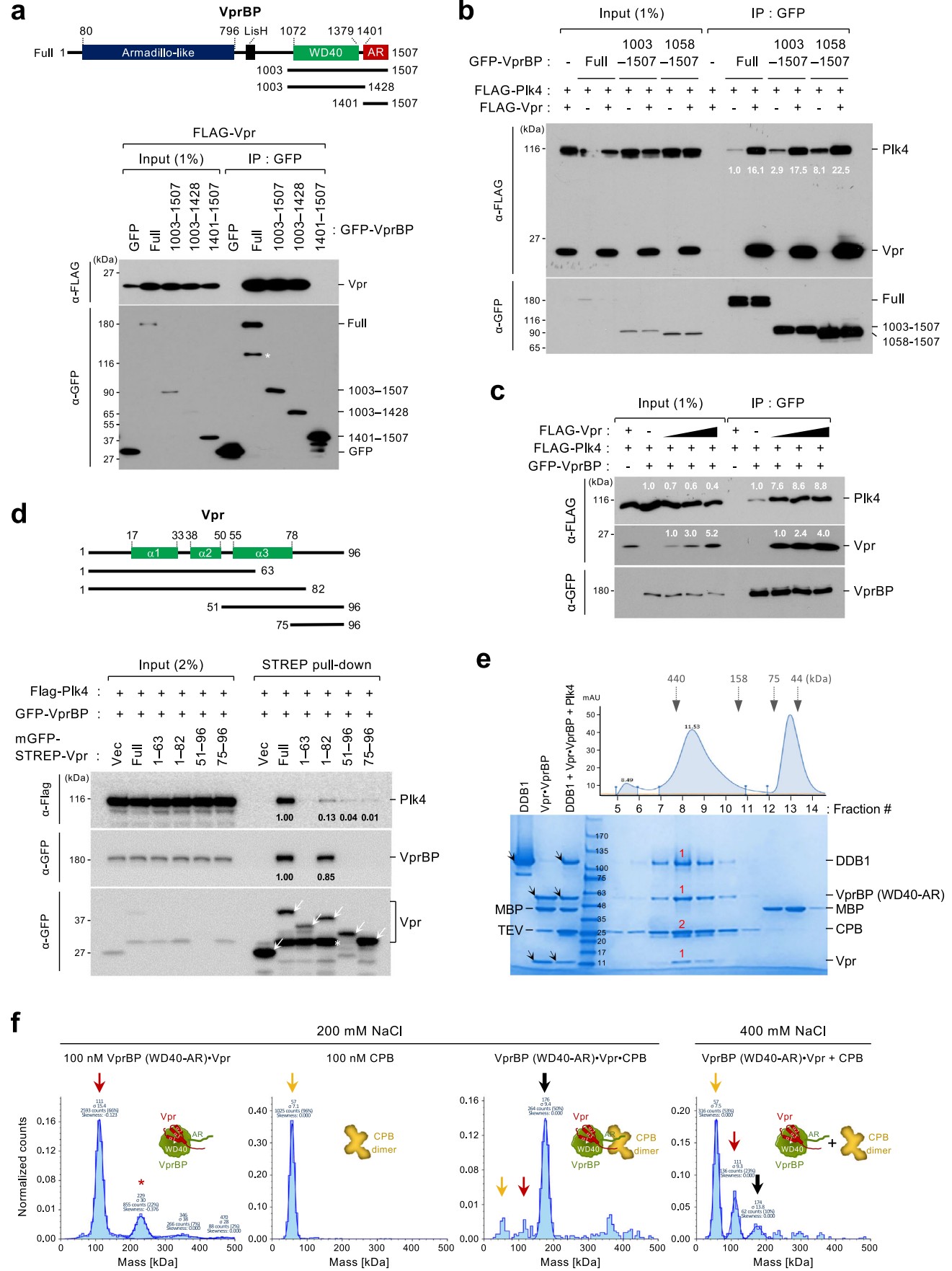

**Fig. 3 | Vpr forms a ternary complex with VprBP WD40-AR and Plk4 CPB.** Coimmunoprecipitation analyses performed with HEK293T cells transfected with the indicated constructs. Asterisk in (**a**), degradation product; Black triangles in (**c**), Increased amounts of Vpr DNA transfected; arrows in (**d**), mGFP-STREP-containing ligands; numbers in (**b**–**d**), relative signal intensities. **e** SEC profile and SDS-PAGE analysis demonstrating the purification of the DDB1•Vpr•VprBP WD40-AR(1057–1507)•Plk4 CPB(581–808) complex. Arrows, respective proteins loaded in each lane; red numbers, an estimated binding stoichiometry approximated from the Coomassie Brilliant Blue–stained protein intensity. Note that the level of coprecipitating VprBP is proportional to that of Vpr expressed in the lysates. **f** Interferometric scattering mass spectrometry (iSCAMS) data showing the Vpr•VprBP WD40-AR•Plk4 CPB complex forming within 1 min after mixing all components at 200 mM NaCl. The complex is sensitive to 400 mM NaCl (the 4th panel). The 11-kDa Vpr, which binds tightly to VprBP, cannot be detected due to its small particle size. Red arrow, the Vpr•VprBP WD40-AR complex (red asterisk, a presumed dimer); yellow arrow, Plk4 CPB; thick black arrow, the Vpr•VprBP WD40-AR•Plk4 CPB complex.

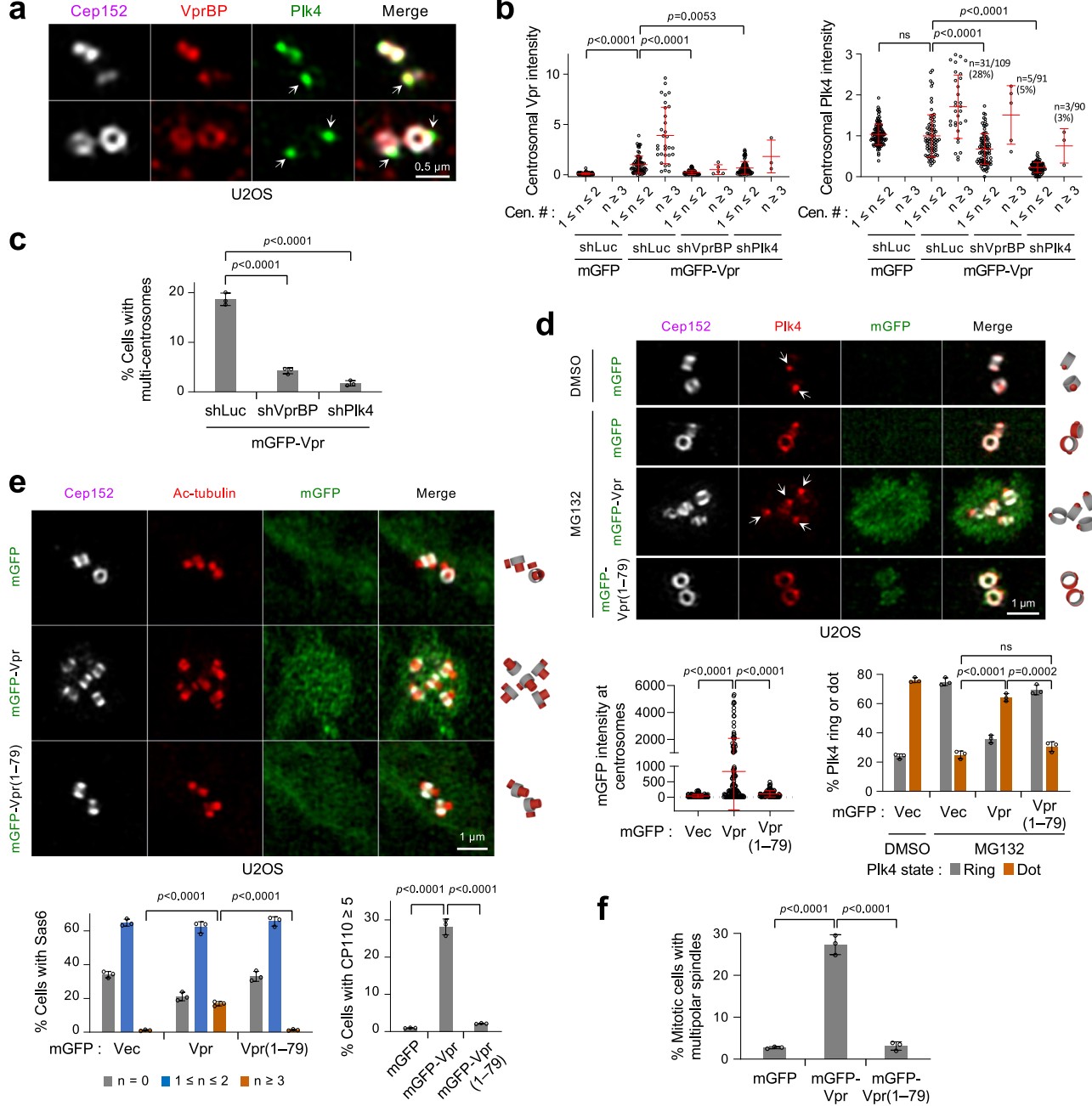

rescued this defect (Supplementary Fig. 4b). This suggests that, albeit at a low degree, the AR-dependent VprBP function contributes to the capacity of Plk4 to dynamically relocate to the procentriole assembly site (the dot state in Supplementary Fig. 4b) from around a centriole (the ring state in Supplementary Fig. 4b).

In a related experiment carried out with U2OS cells stably expressing mGFP-Vpr using a lentiviral system (i.e., transduced cells were selected and maintained under selection conditions throughout the experiment; see Methods), depletion of either VprBP or Plk4 drastically reduced centrosome-localized Vpr signals (Fig. 4b, left, and Supplementary Fig. 4c). This aligns with the observation that Vpr interacts with VprBP and Plk4 (Fig. 3 and Supplementary Fig. 3a–g). VprBP depletion also significantly lowered the level of Plk4 recruited to centrosomes (Fig. 4b, right). This

**Fig. 4 | Vpr enforces a dot-state Plk4 and centriole overduplication in its CT-dependent manner. a** 3D-SIM images showing the localization pattern of VprBP in U2OS cells. Arrows, dot-state Plk4 signals. **b, c** Quantification of centrosome-localizing Vpr and Plk4 signals (**b**) and Vpr-induced, over-duplicated centrosomes (**c**) after treating the mGFP-Vpr-expressing U2OS cells with the indicated shRNA. Confocal images used for quantifications in (**b, c**) are provided in Supplementary Fig. 4c. Data were obtained from three independent experiments. For (**b**), $n = 126$ for shLuc/mGFP ($n \geq 42$/experiment); $n = 109$ for shLuc/mGFP-Vpr ($n \geq 35$/experiment); $n = 91$ for shVprBP/mGFP-Vpr ($n \geq 30$/experiment); $n = 90$ for shPlk4/mGFP-Vpr ($n = 30$/experiment). Bars, mean of $n \pm$ s.d. The numbers and percentages indicate the fraction of cells with more than three centrosomes. For (**c**), $n = 1528$ for shLuc ($n \geq 480$/experiment); $n = 1342$ for shVprBP ($n \geq 441$/experiment); $n = 1389$ for shPlk4 ($n \geq 447$/experiment). Bars, mean of three experiments ± s.d.; $P$ values, unpaired two-tailed $t$-tests. **d** 3D-SIM images obtained from the cells in Supplementary Fig. 4h. Cells were treated with MG132 for 3 h to enrich the ring-state Plk4, as demonstrated previously[43]. MG132 treatment did not alter the cell cycle (Supplementary Fig. 4o). Arrows, dot-state Plk4. Schematic diagrams (right) show the localization patterns of Cep152 and Plk4 signals. Quantified data (graphs) were obtained from three independent experiments. For mGFP intensities, $n = 113$ for mGFP ($n \geq 37$/experiment); $n = 126$ for mGFP-Vpr ($n \geq 41$/experiment); $n = 96$ for mGFP-Vpr(1–79) ($n \geq 30$/experiment). Bars, mean of $n \pm$ s.d. For Plk4 ring or dot quantification, $n = 665$ for mGFP/DMSO ($n \geq 219$/experiment); $n = 494$ for mGFP/MG132 ($n \geq 150$/experiment); $n = 692$ for mGFP-Vpr/MG132 ($n \geq 180$/experiment); $n = 411$ for mGFP-Vpr(1–79)/MG132 ($n \geq 131$/experiment). Bars, mean of three experiments ± s.d.; $P$ values, unpaired two-tailed $t$-tests; ns not significant. **e** Representative 3D-SIM images for cells immunostained with an anti-acetylated-tubulin antibody (See uncropped images in Supplementary Fig. 4j) and quantification of Sas6 and CP110 signals (graphs) using images shown in Supplementary Fig. 4k. Schematic diagrams (right) are shown for Cep152 and acetylated tubulin signals. Quantification was performed from three independent experiments. For Sas6 counts, $n = 1421$ for mGFP ($n \geq 407$/experiment); $n = 1332$ for mGFP-Vpr ($n \geq 411$/experiment); $n = 1351$ for mGFP-Vpr(1–79) ($n \geq 418$/experiment). For CP110 counts, $n = 1369$ for mGFP ($n \geq 417$/experiment); $n = 1418$ for mGFP-Vpr ($n \geq 453$/experiment); $n = 1534$ for mGFP-Vpr(1–79) ($n \geq 471$/experiment). Bars, mean of three experiments ± s.d.; $P$ values, unpaired two-tailed $t$-tests. **f** Quantification of the cells generated in Supplementary Fig. 4h and immunostained with anti-α-tubulin and anti-γ-tubulin antibodies (Representative images shown in Supplementary Fig. 4l) was performed from three independent experiments. $n = 364$ for mGFP ($n \geq 108$/experiment); $n = 380$ for mGFP-Vpr ($n \geq 105$/experiment); $n = 345$ for mGFP-Vpr(1–79) ($n \geq 101$/experiment)]. Bars, mean of three experiments ± s.d.; $P$ values, unpaired two-tailed $t$-tests.

observation would not be expected if VprBP were to promote Plk4 degradation. Not surprisingly, both VprBP and Plk4 were required for overexpressed Vpr to induce multiple centrosomes (judged by the Cep152 signals, which colocalize with γ-tubulin signals throughout the cell cycle) (Fig. 4c and Supplementary Fig. 4c). On the other hand, depletion of Cep78, known to interact with Plk4[58] or VprBP[28] (Supplementary Fig. 4d), failed to influence Vpr's ability to induce multiple centrosomes (Supplementary Fig. 4e). This observation suggests that Cep78 is not required for Vpr•VprBP•Plk4 to generate multiple centrosomes. The Vpr (K27M) mutant relieved of a G2 arrest[59] exhibited an undiminished capacity to bind to VprBP and Plk4 and induce multiple signals of a centriolar scaffold, Sas6[60] (Supplementary Fig. 4f). Thus, Vpr can induce multiple centrosomes independently of Vpr-induced G2 arrest, as reported previously[16]. Unlike HIV-1 Vpr (NL4-3) and Vpr (89.6), HIV-2 Vpr (54% sequence identity with HIV-1 Vpr) induced multiple centrosomes only at a low level, even though it was expressed at a higher level than the HIV-1 Vpr proteins (Supplementary Fig. 4g). This observation is consistent with its inability to form a cooperative complex with VprBP and Plk4 (Supplementary Fig. 3g).

Next, to determine whether Vpr alters Plk4's localization dynamics and Plk4-dependent centriole biogenesis, we performed comparative immunostaining analyses using cells expressing a monomeric GFP (mGFP)-Vpr or mGFP-Vpr(1–79) lacking the Plk4-binding C-terminal 17 residues (Supplementary Fig. 4h). Notably, mGFP-Vpr alone effectively localized to the pericentriolar material (PCM) region, often showing a "nebulous" appearance that encompasses multiple dot-like Plk4 signals (Fig. 4d and Supplementary Fig. 4i). In contrast, the Vpr(1–79) mutant localized poorly to centrosomes and failed to significantly induce the dot-like Plk4 signals (Fig. 4d and Supplementary Fig. 4i). Consistent with this finding, a substantial fraction of cells expressing Vpr, but not the Vpr(1–79) mutant, exhibited multiple acetylated tubulins (Fig. 4e and Supplementary Fig. 4j) and significantly increased the number of Sas6[60,61] and CP110 signals[29,30] (Fig. 4e, graphs, and Supplementary Fig. 4k). Consistent with the data obtained from the Vpr (K27M) mutant (Supplementary Fig. 4f), mGFP-Vpr induced multiple centrioles (Fig. 4d,e and Supplementary Fig. 4i–k) that lead to the formation of multiple centrosomes (Fig. 4f and Supplementary Fig. 4l) independently of the cell cycle (Supplementary Fig. 4m–o). These data suggest that Vpr CT-dependent interaction with Plk4 heightens Plk4's ring-to-dot relocalization and centriole overduplication, ultimately generating multiple centrosomes.

## Vpr enhances Plk4 catalytic activity in vitro and in vivo

Since Plk4's ring-to-dot relocalization is promoted by its *trans*-autophosphorylation activity[41], we examined whether Vpr can influence the catalytic activity of Plk4. In an in vitro kinase assay performed with a glutathione-*S*-transferase (GST)-fused Plk4(1–836; ΔPB3), Vpr•VprBP, but not Vpr(1–79)•VprBP, augmented Plk4's ability to phosphorylate its activational SSTT (i.e., S698, S700, T704, and T707) motif[41,42] by approximately 1.8-fold (Fig. 5a and Supplementary Fig. 5a). This finding suggests that Vpr can promote Plk4 catalytic activity by directly binding to its CPB as demonstrated in Supplementary Fig. 3i.

In cultured cells, Vpr, but not the Plk4 binding-defective Vpr(1–79), generated cytosolic assemblies with coexpressed Plk4 and endogenous VprBP (Supplementary Fig. 5b). Since centrosome-associated Vpr effectively recruits downstream components, such as Sas6 and CP110 (Fig. 4e), we examined whether Vpr can induce Plk4-dependent STIL S1108 phosphorylation, a critical event for recruiting Sas6 to a procentriole assembly site[43]. Our results showed that, under the conditions where mGFP-Vpr and mGFP-Vpr(1–79) were comparably expressed (Supplementary Fig. 5c), mGFP-Vpr augmented the level of Plk4-dependent STIL S1108 phosphorylation by 2.2-fold (Fig. 5b, c). In contrast, consistent with the in vitro data shown in Fig. 5a, the mGFP-Vpr(1–79) mutant only weakly promoted the generation of the p-S1108 epitope (compared with the mGFP control). This finding aligns with a low mGFP-Vpr(1–79) signal level colocalized with Plk4 (Fig. 5b, c right two panels). The level of the STIL p-S1108 signal was largely proportional to the amount of Plk4 present in the Vpr-associated assemblies (Fig. 5c, 1st panel).

## Both Vpr CT and VprBP AR are critical for Vpr•VprBP•Plk4-mediated centriole overduplication

To corroborate the formation of multiple centrioles in mGFP-Vpr-expressing cells, we performed transmission electron microscope (TEM) tomography as described in the Method. The result confirmed the presence of multiple centrioles, often found in a clustered region (Fig. 6a and Supplementary Fig. 6a). This observation is in line with the data shown in Fig. 4b–e.

Since Vpr and VprBP are required for inducing multiple centrioles (Fig. 4c), we then investigated the significance of forming a ternary Vpr•VprBP•Plk4 complex in Fig. 3 in deregulating centriole duplication by immunostaining U2OS cells stably expressing the indicated Vpr and/or VprBP constructs. Under the conditions where exogenous Vpr and VprBP were expressed at comparable levels, Vpr robustly induced centrosome overduplication and VprBP further augmented it (Fig. 6b

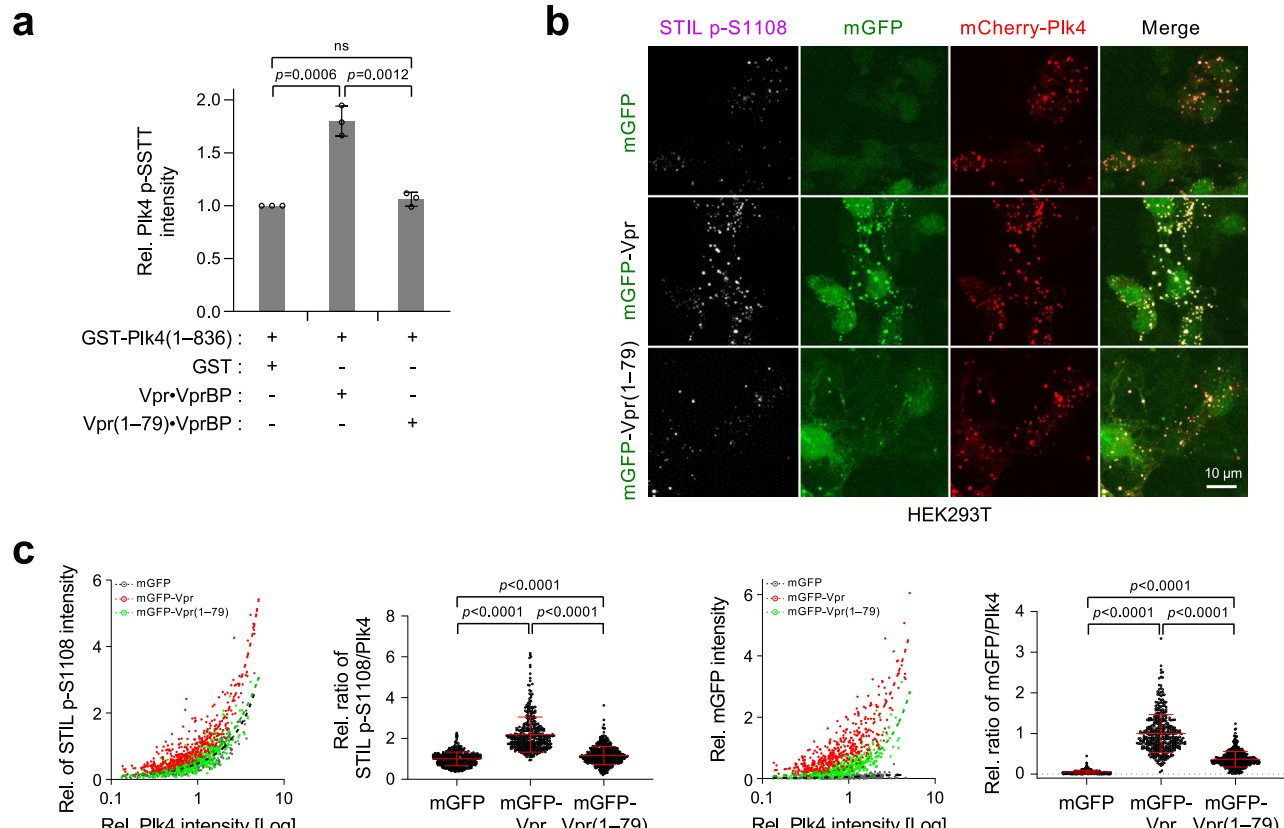

**Fig. 5 | Vpr augments Plk4 kinase activity in vitro and in vivo. a** Quantification of Plk4 *trans*-autophosphorylation activity determined by performing in vitro kinase assays with Plk4(1–836; ΔPB3) in the presence of Vpr•VprBP or Vpr(1–79)•VprBP. The *trans*-autophosphorylated p-SSTT motif in Plk4 was detected by an anti-p-SSTT antibody[41,42]. Data were obtained from three independent experiments. Bars, mean of three experiments ± s.d.; *P* values, unpaired two-tailed *t*-tests; ns not significant. Representative gels used for the quantification are provided in Supplementary Fig. 5a. **b** Representative confocal images showing immunostained HEK293T cells cotransfected with mCherry-Plk4 and mGFP-Vpr or mGFP-Vpr(1–79) for 12 h. STIL p-S1108

antibody was used in the presence of non-phospho-epitope peptide. **c** Quantification of STIL p-S1108 and mGFP fluorescence signals (1st and 3rd panels) and determination of ratios of STIL p-S1108/Plk4 and mGFP-Vpr/Plk4 signal intensities (2nd and 4th panels) performed using the images obtained in (**b**). Interpolated curves (left) were drawn by fitting with nonlinear regression. All the data obtained from three independent experiments are plotted. Per experiment, $n \geq 116$ for mGFP (total $n = 407$); $n \geq 107$ for mGFP-Vpr (total $n = 412$); $n \geq 149$ for mGFP-Vpr(1–79) (total $n = 469$). Bars, mean $n \pm$ s.d.; *P* values, unpaired two-tailed *t*-tests. Immunoblots showing the expression levels of Vpr and Vpr(1–79) are provided in Supplementary Fig. 5c.

and Supplementary Fig. 6b). Both Vpr(1–79; ΔCT) and VprBP(1–1427; ΔAR) mutants defective in Plk4 binding failed to promote this event (Fig. 6b and Supplementary Fig. 6b). Thus, Vpr cooperates with VprBP to induce centrosome overduplication in a manner that requires the Plk4-binding Vpr CT and VprBP AR regions.

Since DDB1 interacting with VprBP coprecipitated with CPB and Plk4 (Fig. 2a and Supplementary Fig. 2c), we examined whether DDB1 or its upstream CUL4A, a cytoplasm-localized cullin family member of ubiquitin ligases[62], is required for Vpr-induced centrosome overduplication. In immunostaining analyses performed with cells expressing mGFP-Vpr, depletion of either CUL4A or DDB1 did not significantly influence the degree of mGFP-Vpr-induced centrosome overduplication (Fig. 6c and Supplementary Fig. 6c). These data suggest that the Vpr•VprBP•Plk4 complex induces centrosome overduplication independently of the VprBP-mediated E3 ligase activity (Fig. 6d and Supplementary Fig. 3j–m).

### HIV-1 Vpr, but not the Plk4 binding–defective Vpr(1–79) mutant, induces centrosome overduplication in various CD4+ cells

We examined the effect of Vpr expression on centriole duplication in multiple CD4+ cells infected with HIV-1 pseudoviruses (see Methods). In TZM-bl cells (derived from HeLa cells) or CEM-SS cells (derived from CEM CD4+ T cells), neither HIV-1 wild type (WT) nor its respective Vpr mutants noticeably altered the levels of various components critical for centriole biogenesis (Supplementary Fig. 7a, b). Under these

conditions, HIV-1 WT drastically induced cells with multiple centrosomes (i.e., greater than two Cep152 signals counted among the p24+ population) (Fig. 7a and Supplementary Fig. 7a, b) by promoting Plk4 relocalization from a ring state to a dot state (Supplementary Fig. 7c), as demonstrated in Fig. 4d. In contrast, its respective HIV-1 Vpr(1–79) and Vpr(-) mutants induced multiple centrosomes only marginally. These data, which corroborate the results obtained from U2OS cells (Fig. 4b–e), suggest that eliminating Vpr CT-dependent Plk4 interaction is sufficient to abolish Vpr-induced centrosome amplification in CD4+ cells.

In a related experiment, HIV-1 Vpr-induced, overduplicated centrosomes were nearly annihilated by the depletion of VprBP (Fig. 7b and Supplementary Fig. 7d). Treatment of HIV-1-infected CEM-SS cells with centrinone, a Plk4-specific inhibitor[63], significantly diminished the level of HIV-1-induced centrosome overduplication (Supplementary Fig. 7e). In addition, immunoprecipitation of Plk4 from HIV-1-infected CEM-SS cells coprecipitated VprBP and Vpr (Fig. 7c). Similar results were obtained with HEK293 cells transduced with Vpr-expressing lentiviruses (Supplementary Fig. 7f). These findings reinforce our notion that HIV-1-induced Vpr•VprBP•Plk4 complex drives centrosome amplification and that VprBP serves as the critical scaffold that enables Vpr to promote Plk4-mediated centriole duplication. Remarkably, while treatment of CEM-SS cells with 2 μM of raltegravir, a well-characterized HIV-1 integrase inhibitor[64], diminished HIV-1-induced multiple centrosomes, it did so to approximately 50% of untreated

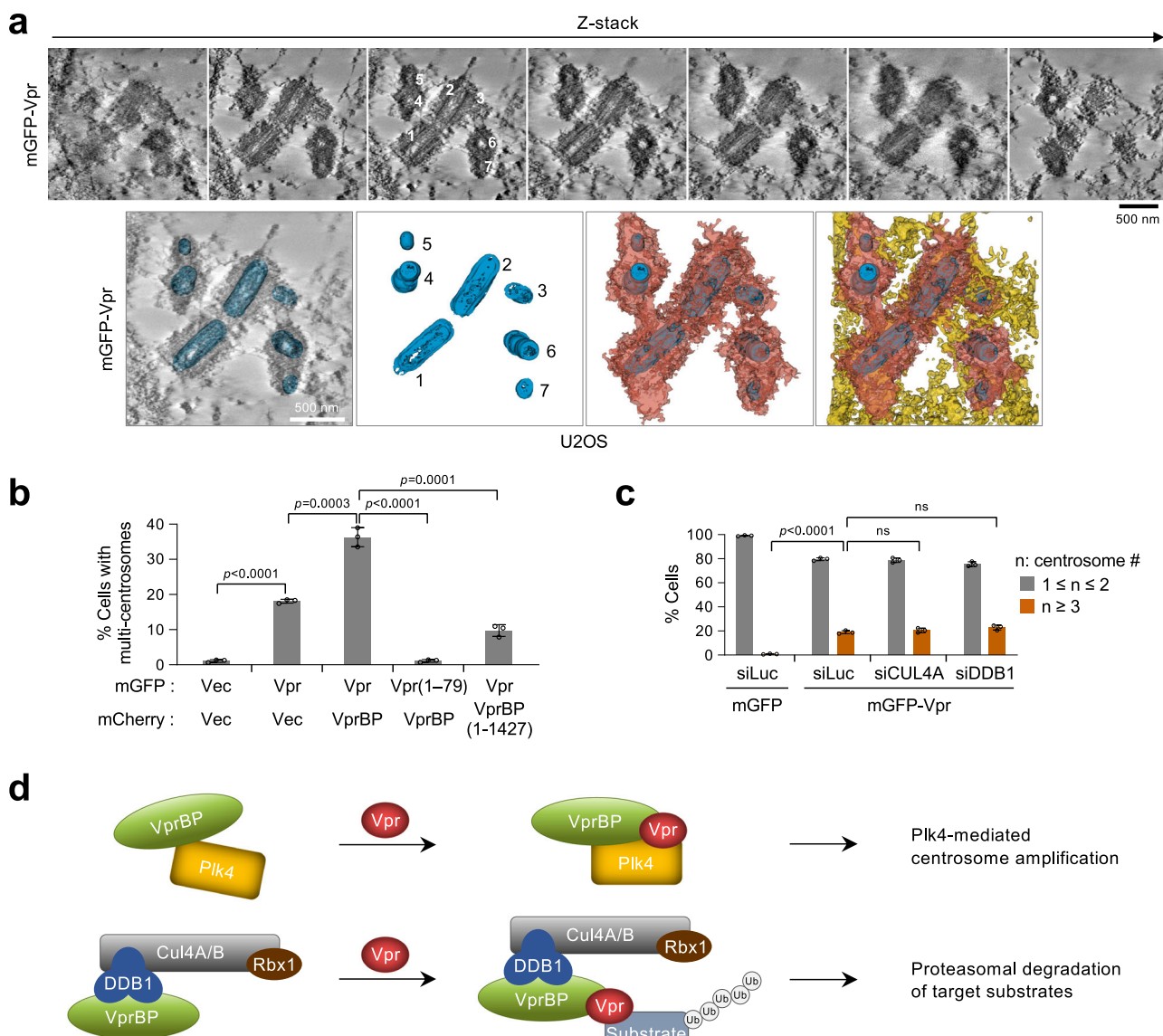

**Fig. 6 | Vpr CT and VprBP AR are required for Vpr•VprBP•Plk4-mediated centriole overduplication independently of the CUL4-DDB1 ubiquitin ligase.**
**a** Tomographic reconstruction and segmentation of centrioles for an asynchronously growing U2OS cell expressing mGFP-Vpr. Regions of the tomogram reconstruction are provided at different z-slices showing mGFP-Vpr-induced multiple centrioles (numbered from 1 to 7). Centrioles (blue) and their directly associated (red) and surrounding (yellow) cellular materials are shown. **b** Quantification of immunostained cells in Supplementary Fig. 6b, stably expressing the indicated Vpr and VprBP constructs using a lentiviral system. Results were obtained from three independent experiments [per experiment, $n \geq 149$ for mGFP + mCherry (total $n = 459$); $n \geq 166$ for mGFP-Vpr + mCherry (total $n = 530$); $n \geq 143$ for mGFP-Vpr + mCherry-VprBP (total $n = 453$); $n \geq 143$ for mGFP-Vpr(1–79) + mCherry-VprBP (total

$n = 446$); $n \geq 134$ for mGFP-Vpr + mCherry-VprBP(1–1427) (total $n = 439$)]. Bars, mean of three experiments ± s.d.; $P$ values, unpaired two-tailed $t$-tests.
**c** Quantification of immunostained cells in Supplementary Fig. 6c after depletion of CUL4A, the cytoplasm-localized form[62], or DDB1 by RNAi. Quantification was performed from three independent experiments [per experiment, $n \geq 609$ for mGFP/siLuc (total $n = 2132$); $n \geq 672$ for mGFP-Vpr/siLuc (total $n = 2174$); $n \geq 624$ for mGFP-Vpr/siCUL4A (total $n = 1933$); $n \geq 623$ for mGFP-Vpr/siDDB1 (total $n = 1927$)]. Bars, mean of three experiments ± s.d.; $P$ values, unpaired two-tailed $t$-tests; ns not significant. **d** Schematic diagrams illustrating how Vpr hijacks two distinct cellular complexes—one that overdrives the Plk4-dependent centriole duplication event (top) and the other that promotes the VprBP-mediated E3 ligase activity (bottom).

control cells (Supplementary Fig. 7g, h). This suggests that Vpr encapsidated in the virions is sufficient to induce a significant level of multiple centrosomes.

Next, we investigated whether HIV-1 Vpr CT-dependent centrosome amplification is sufficient to induce aneuploidy, a condition that could drive oncogenesis[10,44–46]. To this end, purified CD4[+] T cells from healthy individuals were stimulated on an anti-CD3 and anti-CD28 antibody–coated plate for one day, infected with HIV-1 WT or its respective HIV-1 Vpr(1–79) mutant for 8 h, and additionally cultured for four days before analyses. In line with the data described above, the primary CD4[+] T cells infected with HIV-1 WT, but not the

Vpr(1–79) mutant, exhibited centrosome amplification (marked by Cep152, Cep192, acetylated tubulin, and CP110 signals) in approximately 19% of the p24[+] population (Fig. 7d and Supplementary Fig. 7i, j). Consistently, metaphase chromosome spread analyses revealed that, when compared to the uninfected control cells, the cells infected with HIV-1 WT, but not the Vpr(1–79) mutant, increased the fraction of aneuploid cells by approximately 20% (mean of three experiments carried out with the primary CD4[+] T cells purified from as many healthy subjects) (Fig. 7e and Supplementary Fig. 7k). Considering that approximately 10% of primary CD4[+] T cells are typically p24[+] under our experimental conditions, the rate of generating

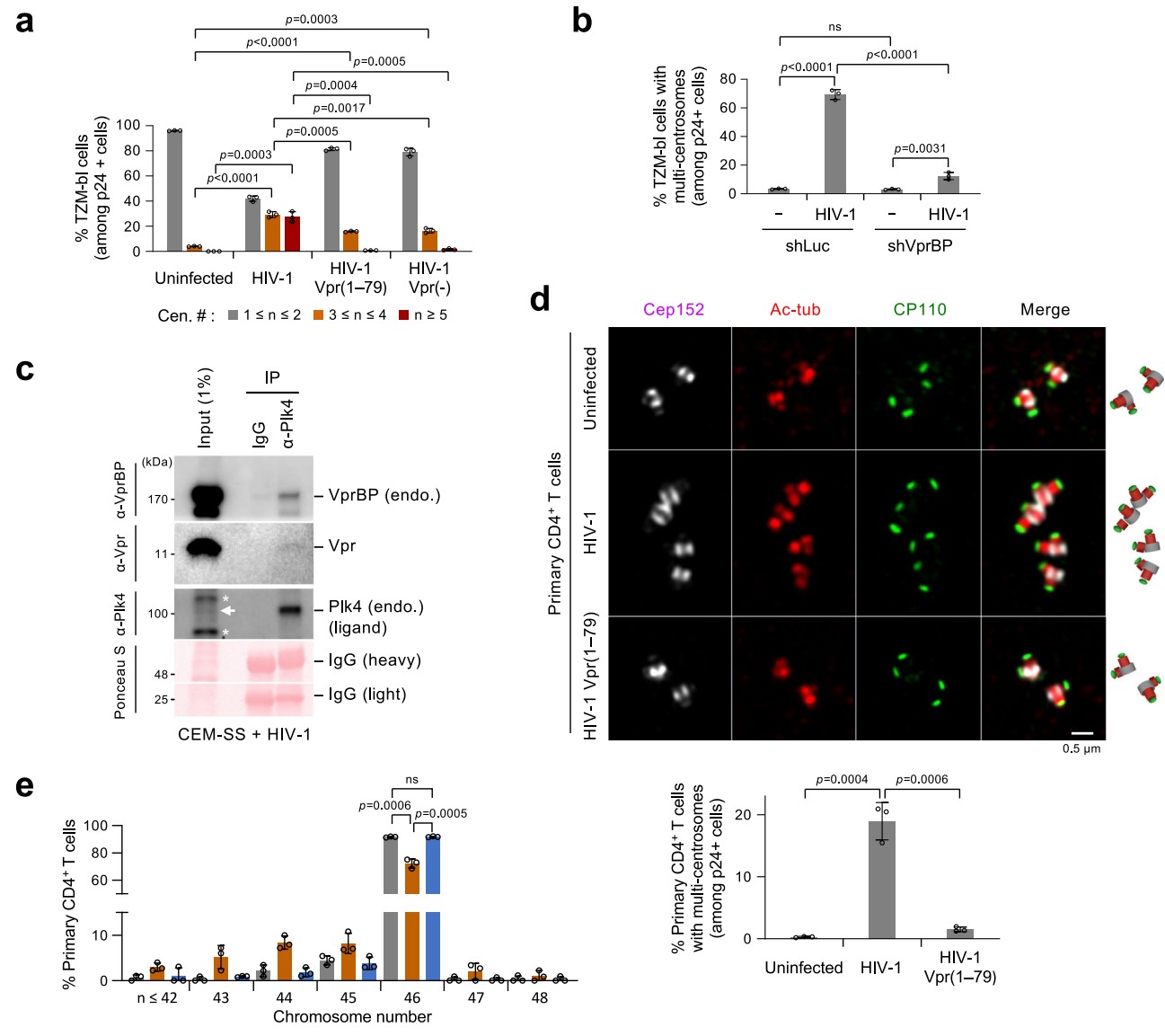

**Fig. 7 | Induction of centrosome amplification and aneuploidy by HIV-1 WT but not the Plk4 binding–defective HIV-1 Vpr(1–79) mutant. a** Quantification of immunostained CD4⁺ TZM-bl cells infected with the indicated HIV-1 WT or Vpr mutants (representative images shown in Supplementary Fig. 7a). Centrosomes (marked by Cep152 signals) were counted among the HIV-1-infected p24⁺ cells. HIV-1 Vpr(-) denotes a Vpr-null mutant. Results were quantified from three independent experiments [per experiment, $n \geq 213$ for uninfected cells (total $n = 707$); $n \geq 202$ for HIV-1 (total $n = 621$); $n \geq 246$ for HIV-1 Vpr(1–79) (total $n = 795$); $n \geq 204$ for HIV-1 Vpr(−) (total $n = 637$)]. Bars, mean of three experiments ± s.d.; *P* values, unpaired two-tailed *t*-tests. **b** Quantification of cells with multi-centrosomes (i.e., Cep152 signals) performed after infecting them with HIV-1 and silencing them for control luciferase (shLuc) or VprBP (shVprBP). Counts were among the p24⁺ population (representative images provided in Supplementary Fig. 7d) from three independent experiments. Per experiment, $n \geq 232$ for uninfected/shLuc (total $n = 737$); $n \geq 209$ for HIV-1/shLuc (total $n = 656$); $n \geq 222$ for uninfected/shVprBP (total $n = 713$); $n \geq 245$ for HIV-1/shVprBP (total $n = 750$)]. Bars, mean of three experiments ± s.d.; *P* values, unpaired two-tailed *t*-tests; ns not significant. **c** Coimmunoprecipitation and immunoblotting analyses performed with CEM-SS cells infected with HIV-1 and treated with 200 nM of centrinone for 3 h to increase Plk4 stability[56]. Endogenous Plk4 was coimmunoprecipitated with a mouse anti-Plk4 (6H5) antibody

and then subjected to immunoblotting analyses. The same membrane stained with Ponceau S is provided. Arrow, endogenous (endo.) Plk4; asterisks, cross-reacting proteins. **d** 3D-SIM and quantification of immunostained primary CD4⁺ T cells purified from healthy PBMCs, infected with HIV-1 WT or Vpr(1–79) mutant for 8 h, and cultured on the anti-CD3 and anti-CD28 antibody–coated plate for 4 days. Cells generated in Supplementary Fig. 7i, left (immunoblotting), were immunostained, and obtained images in Supplementary Fig. 7i, right, were cropped to generate Fig. 7d. Schematic diagrams (right) show merged images. Using the cells in Supplementary Fig. 7j, centrosome numbers (marked by Cep152 and Cep192 signals) (graph) were quantified among the p24⁺ population from three independent experiments [per experiment, $n \geq 790$ for uninfected cells (total $n = 2925$); $n \geq 535$ for HIV-1 (total $n = 1996$); $n \geq 520$ for HIV-1 Vpr(1–79) (total $n = 2323$)]. Bars, mean of three experiments ± s.d.; *P* values, unpaired two-tailed *t*-tests. **e** Quantification of chromosome numbers for primary CD4⁺ T cells infected with the indicated viruses and cultured as in (**d**). Results were quantified from three independent experiments performed with CD4⁺ T cells from three healthy individuals. Representative chromosome spread images are provided in Supplementary Fig. 7k. Per experiment, $n \geq 87$ for uninfected cells (total $n = 288$); $n \geq 88$ for HIV-1 (total $n = 299$); $n \geq 97$ for HIV-1 Vpr(1–79) (total $n = 328$). Bars, mean of three experiments ± s.d. ***$P < 0.001$ (unpaired two-tailed *t*-test). ns not significant.

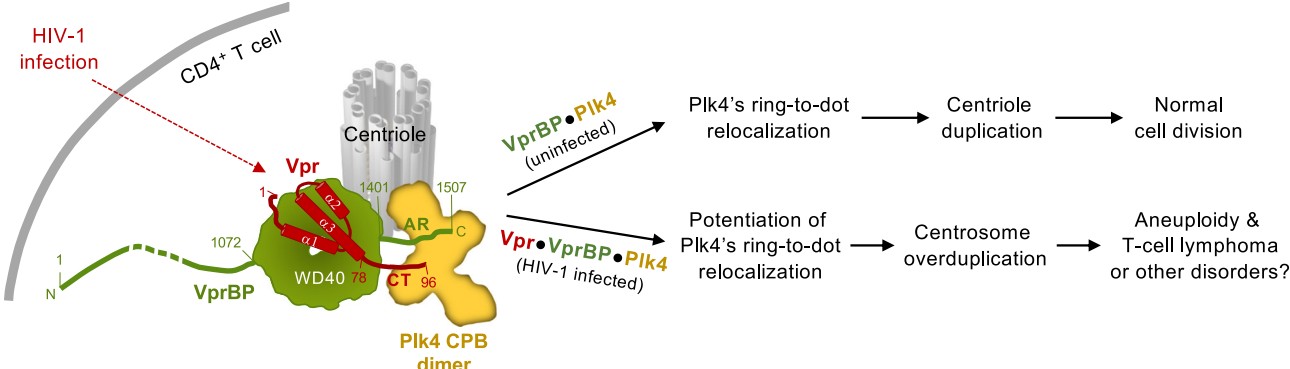

**Fig. 8 | Model illustrating how HIV-1 Vpr generates the Vpr•VprBP•Plk4 complex and induces centrosome amplification and aneuploidy.** Under unperturbed conditions, VprBP binds to Plk4 through its C-terminal AR and contributes to the centriole duplication process by promoting Plk4's ring-to-dot relocalization. Overexpression of catalytically active Plk4 can augment this process and induce multiple procentrioles[30,41]. When HIV-1 infects CD4+ T cells, its accessory protein, Vpr, forms a ternary complex with VprBP and Plk4 and enhances Plk4's catalytic activity. This potentiates Plk4's ability to undergo LLPS-mediated ring-to-dot relocalization and induce centrosome overduplication that can lead to aneuploidy. Given that integration of HIV-1 proviruses into oncogenes, such as STAT3 and LCK, can promote the development of T-cell lymphomas[5], HIV-1 may promote oncogenesis by driving both provirus integration–mediated oncogene activation and Vpr-mediated deregulation of the centriole duplication machinery.

aneuploid cells by HIV-1 appeared higher than expected. This could be in part due to the 2- to 6-fold faster proliferation of CD4+ T cells in subjects with HIV-1[65] with weakened mitotic checkpoint, as suggested previously[66]. Nevertheless, these findings and the data shown in Supplementary Fig. 3i suggest that the Vpr CT-dependent interaction with Plk4 CPB underlies the induction of aneuploidy in these cells.

## Discussion

People living with HIV-1 are at a greatly increased risk of developing non-Hodgkin's lymphoma (NHL)[67,68]. Although not as prevalent as B-cell lymphomas, a study analyzing the AIDS–Cancer Match Registry of more than 300,000 adults with AIDS shows an approximately 15-fold increase in T-cell lymphomas (24-fold increase in the case of peripheral T-cell lymphoma)[69]. Notably, while the incidence of NHL and other AIDS-defining cancers becomes drastically reduced during the ART period, the risk of developing these diseases remains higher when compared with the general population[70]. A large-scale meta-analysis shows that the risk of developing NHL is far greater in individuals with HIV-1 than in immunosuppressed transplant recipients[68]. These findings suggest that factor(s) other than HIV-1-induced immune deficiency promotes HIV-1-associated cancers.

How T-cell lymphomas can arise under conditions where HIV-1 can progressively destroy them remains unclear. A body of evidence suggests that HIV-1 proviruses integrated into several oncogenes, such as signal transducer and activator of transcription 3 (STAT3) and lymphocyte-specific protein tyrosine kinase (LCK), cause clonal expansion of infected T cells, thus paving the way for developing T-cell lymphomas[5,71,72]. In addition, the accumulation of mutations in oncogenes and tumor suppressors may help promote altered cell proliferation[73] by evading cellular responses, such as the Vpr-induced G2 arrest.

Here we showed that centrosomes are amplified in the primary CD4+ T cells of people living with HIV-1, and the level of the amplification is significantly reduced after ART (Fig. 1). Fascinated by the finding, we explored whether an HIV-1-encoded protein(s) can directly deregulate host cellular processes that can contribute to oncogenesis. In this regard, given that aneuploidy is considered a cause of cancer development[45,46,74], our discovery of a ternary Vpr•VprBP•Plk4 complex and the ability of the complex to potentiate Plk4-mediated centriole duplication and induce aneuploidy are significant (Fig. 8).

An intriguing finding is that virion-packaged Vpr (i.e., Vpr encapsidated in the virion) was sufficient to induce a significant level of centrosome overduplication (Supplementary Fig. 7g, h). This suggests that Vpr can efficiently hijack the Plk4-dependent centriole duplication machinery by forming the Vpr•VprBP•Plk4 complex (Fig. 8). Studies show that Vpr is present in the plasma of people living with HIV-1 in the range of 10 pg/mL–10 ng/mL of blood[75–77] and a substantial amount (4 pg/mL) of Vpr still circulates in the blood of ART-treated patients[75]. These findings are in line with the notion that active reservoirs produce viral proteins, such as Vpr during ART treatment[78,79]. In addition, Vpr exhibits membrane-penetrating properties[80,81]. These observations help explain why CD4+ T cells obtained after ART still exhibit 0.6–1.3% (on average, 26% of the cells analyzed before ART) of the population with overduplicated centrosomes (Supplementary Fig. 1d). By extension, Vpr circulating in the bloodstream of a healthy person may give rise to centrosome abnormality–associated tumorigenesis in cells other than CD4+ T cells. Exploring this possibility could be an attractive direction for future research.

Our data showed that Vpr's capacity to interact with both VprBP WD40 and Plk4 CPB is essential to bolster the ability of the VprBP•Plk4 complex to induce centriole biogenesis. This new function of Vpr was unanticipated, given that Vpr and VprBP have been shown to cooperatively promote the degradation of various cellular targets[24–27]. In line with this novel role of Vpr in deregulating Plk4-dependent centriole duplication, depleting the component of cullin-4-RING E3 ubiquitin ligase, CUL4A or DDB1, did not alter the degree of Vpr-induced centrosome overduplication (Fig. 6c and Supplementary Fig. 6c). In addition, Vpr and VprBP failed to influence Plk4 stability in vivo or increase the level of Plk4 ubiquitination in vitro (Supplementary Fig. 3j–m).

How does the Vpr•VprBP•Plk4 complex induce centriole overduplication? When coexpressed, all three proteins proficiently generated colocalized signals in a manner that requires the Vpr CT(80–96) region (Supplementary Fig. 5b). Likewise, the association of Vpr with VprBP and Plk4 was apparent in the coimmunoprecipitation analysis performed with CEM-SS cells infected with HIV-1 (Fig. 7c). In a related experiment, Vpr but not Vpr(1–79) enhanced the ability of Plk4 to trans-autophosphorylate its activational SSTT site in vitro (Fig. 5a and Supplementary Fig. 5a) and significantly (2.2-fold) increased the level of phosphorylated STIL at the S1109 motif in vivo (Fig. 5b, c and Supplementary Fig. 5c). Consistent with this finding, HIV-1 WT, but not the Plk4 binding–defective VprΔ(80–96) mutant, potently induced

centrosome overduplication and aneuploidy in primary CD4$^+$ T cells purified from PBMCs of healthy human subjects (Fig. 7d, e). The approximately two-fold increase in Plk4 activity by Vpr is comparable to the previous observation that a modest elevation of Plk4 level (<2-fold) is sufficient to induce centrosome amplification and aneuploidy that lead to the generation of various spontaneous tumors in a mouse model[48]. Collectively, targeting the Vpr C-terminal motif could be an attractive strategy to antagonize the function of the Vpr•VprBP•Plk4 complex and thereby reduce the occurrence of HIV-1-associated cancers.

Centrosome amplification, a cause of chromosomal instability and aneuploidy[10], is prevalent among hematological malignancies[6–9], and its degree of abnormalities closely correlates with tumor grades[82,83]. As a master regulator of centriole duplication, deregulated Plk4 activity appears to be tightly associated with oncogenesis[10,44,84]. Here, we demonstrated that, through the formation of the Vpr•VprBP•Plk4 complex, Vpr deregulates Plk4's functionality and induces centrosome amplification and aneuploidy in HIV-1-susceptible CD4$^+$ T cells. In light of a recent view that HIV-1 provirus integration into oncogenes could lead to developing T-cell lymphomas[5], this study may offer an unexplored avenue for investigating the molecular mechanism underlying how comorbid cancers arise in people living with HIV-1.

## Methods

### Study participants

The Individuals infected with HIV-1 were recruited as study participants at the National Institutes of Health (NIH), Clinical Center in Bethesda, MD. Leukapheresis products were obtained from the study participants in accordance with clinical protocols (NIH/ IRB FWA00005897) and relevant ethical regulations approved by the Institutional Review Board of the NIH. All study participants provided written informed consent for the research.

### Plasmid constructs

The control vector pHR.J-ZZ-TEV and pHR.J-ZZ-TEV-Plk4(581–884) constructs (pKM4283 and pKM4285) were generated by inserting a HindIII (end-filled)-BamHI or HindIII (end-filled)-XhoI fragment into the pHR'.J-CMV-SV-puro vector[85] digested by SamI and BamHI or SmaI or SalI, respectively. The FLAG-fused Plk4(581–970) construct (pKM4591) is described by Park et al. (*Nat Comm*, 2019,). The various FLAG-fused Plk4 constructs (pKM3445, pKM3448, and pKM3506–pKM3509) and the HA-tagged Plk4 construct (pKM3855) were reported previously[36,37]. The construct was generated similarly by using the pCI-neo-HA vector (pKM1209). The pKM7582 construct was generated by sequentially inserting the EcoRI-AscI fragment of the Halo tag and the AscI-SalI fragment of the Plk4 gene into the corresponding sites of the pHR'.J-CMV-SV-puro (pKM2994) vector. The pKM7089 construct was generated by inserting the Plk4 gene into the pHR'.J-CMV-SV-puro-mCherry (pKM6287) vector digested with BamHI and SalI. The HA-fused VprBP construct (pKM4358) was constructed by inserting a PmeI-XhoI fragment into the pCI-neo-HA vector (pKM1209) at the corresponding sites. Various GFP-fused VprBP clones (pKM4543–pKM4548, pKM5705, pKM5684, pKM5686, pKM5721–pKM5723, pKM5805, pKM4617, pKM4836) were constructed by inserting PmeI-NotI fragments into the pCI-neo-GFP vector (pKM3828) digested by the same enzymes. The constructs expressing VprBP (pKM4630) and its truncates (pKM5284, pKM5412, and pKM5283) were generated by inserting a PmeI-XhoI or PmeI-NotI fragment, respectively, into the pKM2795 vector digested with corresponding enzymes. To generate the lentivirus-based, the mCherry-tagged VprBP full length or VprBP(1–1427) (pKM7574 or pKM7794), the EcoRI-SalI fragments were inserted into the pHR'.J-CMV-SV-puro (pKM2994) digested by the same enzyme.

The FLAG-tagged Vpr (pKM5330) and HA-tagged Vpr WT or its K27M mutant (pKM5523 or pKM5755) were generated by inserting a PmeI-XhoI fragment into the pKM2795 and pKM1209 vector, respectively. Unless otherwise indicated as gift constructs (see Supplementary Table 1), all the Vpr constructs generated in this study are derived from the infectious molecular clone pNL4-3[86]. The open reading frame (ORF) of Vpr in pKM5330 contains Y15, S28, N41, and R85 residues. It is identical in the primary sequence to the Vpr constructs provided as gifts by Jae-Il Park (MD Anderson Cancer Center, TX), Angela M. Gronenborn (University of Pittsburgh, PA; pKM4759), and Jeremy Luban (University of Massachusetts Medical School, MA). The FLAG-Vpr construct (pKM4758; Lai strain containing H15, N28, G41, and Q85 residues) was provided by Michael Emerman (Fred Hutchinson Cancer Center, WA). Both Vpr variants showed similar cooperativity in forming a ternary complex with VprBP and Plk4 (Supplementary Fig. 3c). Various Vpx constructs (pKM4753–pKM4757) were provided by Monsef Benkirane (Institut de Génétique Humaine, Montpellier, France) and Michael Emerman (Fred Hutchinson Cancer Center, WA). The pKM8004 and pKM8006 constructs expressing Vpr (89.6 strain) and HIV-2 Vpr (NWK08 strain) were generated by inserting a PmeI-XhoI fragment containing each gene into the pCI-neo-FLAG vector (pKM2795). The pKM7799 construct was generated by inserting the mGFP-STREP-TEV into the pHR'.J-CMV-SV-puro vector digested by EcoRI and BamHI. Various mGFP-STREP-TEV-fused Vpr clones (pKM7653–pKM7655 and pKM7666–pKM7667) were constructed by inserting AscI-BamHI fragments into the pHR'.J-CMV-SV-puro-mGFP-STREP-TEV vector (pKM7652) digested by the same enzymes. Lentiviral mGFP-Vpr constructs (pKM7618, pKM7792) were generated by inserting an AscI-BamHI fragment containing Vpr WT or Vpr(1–79) into the pHR'.J-CMV-SV-puro-mGFP (pKM7410) vector digested by the corresponding enzymes. The lentiviral pKM8001 and pKM8003 constructs expressing either Vpr (89.6) or HIV-2 Vpr (NWK08) were generated by inserting the respective genes into a pHR'.J-CMV-SV-puro-mGFP vector digested with AscI and BamHI. The lentiviral pKM7632 construct expressing FLAG-STREP-TEV-Vpr was generated using the pHR'.J-CMV-SV-puro-FLAG-STREP-TEV (pKM7631) vector digested by AscI and BamHI.

The HA-fused DDB1 construct (pKM4354) was generated by cloning a PmeI-NotI fragment into the pKM1209 vector digested by the same enzymes. The HA-tagged Cep57, HA-tagged Cep63, and FLAG-tagged Cep152 constructs (pKM1234, pKM1235, and pKM2809) were described previously[55,87]. The constructs, pKM3676 and pKM2805, were generated by inserting Plk1 and Cep78 genes, respectively, into the pCI-neo-FLAG vector digested with PmeI and NotI enzymes. The pKM1226 construct was generated by inserting the SmaI-NotI fragment of Cep78 into the pCI-neo-HA vector digested with EcoRV and NotI enzymes. The lentiviral pKM7084 construct expressing STIL was generated by inserting a AscI-PmeI fragment of STIL into the pHR'.J-CMV-SV-puro-FLAG vector digested by the same enzymes.

pLKO.1 hygro-shLuc, hygro-shPlk4, and hygro-shVprBP (pKM7743, pKM7744, and pKM7746) were generated by inserting an AgeI-EcoRI fragment into the pLKO.1 hygro vector (Addgene, Watertown, MA).

An *env*-deleted variant of pNL4-3, pNLenv1 (pKM7595), was generated as follows: An EcoRI/BamHI fragment from pNL43 was subcloned into pUC18. The plasmid was then digested with KpnI and BglII, the overhanging ends filled-in with DNA polymerase and the plasmid religated. This operation created a 1264 bp out-of-frame deletion in the *env* gene. The shortened EcoRI/BamHI fragment was then cloned back into pNL4-3, resulting in pNLenv1. To create a Vpr(-) variant of pNLenv1 (pKM7628), a SphI/BamHI fragment containing parts of *gag*, all of *pol* and *Vif*, and parts of *vpr* genes was subcloned into pUC18. A stop codon was then introduced into the vpr open reading frame after residue 2 using PCR-based mutagenesis. This operation did not affect the overlapping *vif* gene. The mutated fragment was then cloned back into the pNLenv1 backbone via the unique PflMI, BamHI restriction

sites. The env-deleted pNL4-3env1 Vpr(1–79) (pKM8247) was constructed by engineering an EcoRI-SalI fragment containing a stop codon after S79 and replacing the EcoRI-SalI fragment in pKM7595. The EcoRI-SalI fragment was generated with two complementary oligonucleotides (5′-AATTCTGCAACAACTGCTGTTTATCCATTTCAGAAT TGGGTGTCGACATAGC**TGA**TG-3′ and 5′-TCGACA**TCA**GCTATGTCGAC ACCCAATTCTGAAATGGATAAACAGCAGTTGTTGCAG-3′, which yield EcoRI and SalI overhangs at each end of the fragment upon annealing them. The stop codon (TGA), which is placed immediately after the S79 residue is shown in boldface type. The Vif and Tat ORFs remain untouched. The construct was confirmed by sequencing.

For protein expression in bacterial cells, the His-MBP-TEV-CPB(581–808) construct (pKM3677) was reported previously[37]. The His-MBP-M-CTD(468–970) (pKM7608) was generated by replacing the Plk4 CPB fragment in pKM3677 with the Plk4 M-CTD(468–970) fragment digested with NdeI and XhoI. The His-MBP-TEV-D5-FLAG-M-CTD (pKM7643), -CTD (pKM7671), or M-CPB (pKM7672) constructs were generated similarly. The His-MBP-TEV-Plk4 WT and its respective K41M mutant (pKM4601 and pKM4602, respectively) were generated similarly to pKM7608 by inserting the corresponding fragments at the NdeI and XhoI sites. The pETDuet-1-His-MBP-TEV-based construct, dually expressing VprBP(1446–1507) and CPB(581–808) (M6934), was generated by inserting respective PmeI-NotI and NdeI-XhoI fragments into the corresponding enzyme sites. The NusA-Vpr-His (pKM6931) and UNG2 were described previously[27,49].

The pTriEx-4 (Addgene)-based insect construct dually expressing His-MBP-TEV-VprBP(1057–1507) and Vpr (pKM7669) was generated by inserting the entire DNA sequence synthesized by GenScript Biotech (Piscataway, NJ) at the AscI and XhoI sites. An additional SV40 poly(A) sequence, a p10 promoter, and a ribosome-binding site sequence were inserted before the Vpr ORF to ensure independent Vpr expression. The insect construct expressing DDB1 was reported previously[49]. The constructs, pKM7731 and pKM7860, were generated similarly by inserting the entire DNA synthesized by GenScript Biotech.

All the constructs used for this study are summarized in Supplementary Table 1.

## Cell culture

U2OS cells (for imaging analyses) and HEK293T cells (for lentivirus production and coimmunoprecipitation analysis) were cultured as recommended by the American Type Culture Collection (ATCC). Two CD4$^+$ cell lines (TZM-bl [ARP-8129; contributed by John C. Kappes and Xiaoyun Wu] and CEM-SS [ARP-776; contributed by Peter L. Nara]) were obtained through the NIH HIV Reagent Program, Division of AIDS, National Institute of Allergy and Infectious Diseases, NIH. U2OS cells were cultured in McCoy's 5 A (Invitrogen), HEK293T and TZM-bl cells were cultured in Dulbecco's Modified Eagle Medium (Thermo Fisher Scientific), and CEM-SS and T cells were cultured in RPMI 1640 Medium (Thermo Fisher Scientific). All media were supplemented with 10% fetal bovine serum (FBS). Where indicated, cells were treated with cycloheximide (100 µg/mL) to examine the half-life of cellular proteins after blocking their translational elongation capacity. In addition, cells were treated with 10 µM of MG132, a reversible proteasome inhibitor[88], for 3 or 6 h, as indicated, to stabilize cellular proteins by inhibiting the degradation of ubiquitin-conjugated proteins. To inhibit Plk4 catalytic activity, CEM-SS cells were treated with 200 nM of centrinone[63] for 48 h before analysis. To inhibit the HIV-1 integrase, CEM-SS cells infected with HIV-1 were treated with the indicated concentrations of raltegravir for 38 h.

To isolate human CD4$^+$ T cells, apheresis from healthy individuals was collected from the NIH blood bank. Cells were layered on Ficoll-Paque premium (Fisher Scientific) and centrifuged at $600 \times g$ for 20 min at room temperature. The buffy coat containing PBMCs was collected and washed, and CD4$^+$ T cells were purified using the EasySep human CD4$^+$ T-cell isolation kit (STEMCELL Technologies). T cells were

then stimulated by culturing them on anti-CD3 (2 µg/mL) and anti-CD28 (2 µg/mL) antibody–coated plates in complete RPMI 1640 Medium supplemented by 10% FBS.

Sf9 insect cells were cultured in Grace's Insect Medium (Thermo Fisher Scientific) supplemented with 10% FBS, 1% Antibiotic-Antimycotic (Thermo Fisher Scientific), and 0.1% Pluronic F-68 (Sigma-Aldrich). Exponentially growing cells were incubated in a temperature-controlled orbital shaker at 150 rpm and 27 °C. For protein expression, Tni-FNL (FNL Hi5) cells were cultured in Sf-900 III SFM (Thermo Fisher Scientific) and shaken at 150 rpm at 27 °C. Cells were passaged when their counts were $>4 \times 10^6$ viable cells/mL. After infection with viruses, the cells were cultured for additional 2.5 days and harvested for protein purification.

## Transfection

For transfection, the calcium phosphate coprecipitation method was used for lentivirus production, while Lipofectamine RNAiMAX (Thermo Fisher Scientific) was used for short interfering RNA (siRNA)-based gene silencing. Transfection using PEI MAX (Polysciences) was implemented to express proteins for immunoprecipitation.

Endogenous VprBP, Plk4, DDB1, CUL4A, βTrCP, or Cep78 were depleted by either transfecting cells with an siRNA targeting a specific gene of interest or transducing them with lentiviruses expressing the respective short hairpin RNA (shRNA). All the siRNAs and shRNAs used for this study are listed in Supplementary Table 2.

BacMagic Transfection Kit (MilliporeSigma) was used for virus production in Sf9 cells according to the manufacturer's protocol.

## Lentivirus and cell line generation

Lentiviruses expressing the gene of interest were produced by cotransfecting HEK293T cells with pHR′-CMVΔR8.2Δvpr, pHR′-CMV-VSV-G (protein G of vesicular stomatitis virus), and the respective pHR′.J-CMV-SV-puro- or pLKO.1 hygro-based constructs listed in Supplementary Table 1. The resulting viruses were used as described previously[89]. Where indicated, cells were additionally transduced with lentiviruses expressing shRNAs (listed in Supplementary Table 2) to deplete the respective endogenous proteins before further analysis. To stably maintain the expression of lentivirus-encoded constructs and/or depletion of target proteins, cells were continuously cultured under puromycin (2 µg/mL) and/or hygromycin (300 µg/mL) during the entire experimental period.

## Pseudo-HIV-1 production and infection

Pseudo-HIV-1 WT and Vpr mutants were produced as described previously[90]. In short, HEK293T cells were cotransfected with pNL4-3env1, pNL4-3env1 Vpr(1–79), or pNL4-3env1 Vpr(-) with pCMV-VSVG at a 9:1 ratio. Forty-eight hours after transfection, viruses were collected and kept frozen until use.

TZM-bl and CEM-SS cells were infected with the viruses above for 12 h and the resulting cells were cultured in a fresh medium for 2.5 days (for TZM-bl) or 1.5 days (for CEM-SS) before immunostaining and immunoblotting analyses.

Primary CD4$^+$ T cells purified from healthy individuals as described above were infected 24 h after stimulating the cells on an anti-CD3 (2 µg/mL) and anti-CD28 (2 µg/mL) antibody–coated plate. After removing supernatant viruses 8 h postinfection, the resulting cells were plated again on the antibody-coated plate and cultured for four days before subjecting them to immunostaining and metaphase chromosome spread analyses. Immunostaining analysis with anti-p24 antibody showed that approximately 10% of T cells were typically infected under our experimental conditions.

The multiplicity of infection (MOI) was determined using an HIV-1 p24 ELISA kit (Abcam) and TZM-bl cells. The MOI was calculated as reported previously[91]. Under our experimental conditions, TZM-bl and CEM-SS cells were typically infected at the MOI of 1. Under these

conditions, TZM-bl cells showed approximately 80% of p24⁺ cells. All the viruses used in this study are replication-deficient, and their efficiency of cell entry, which is a stochastic process, differed greatly depending on the cell types used.

In a related experiment, the relative MOI of the HIV-1 WT or the Vpr(1–79) viruses used for this study was determined using the Steady-Glo® luciferase assay system (Promega) to ensure they were used in equal amounts.

To infect primary CD4⁺ T cells, cells were stimulated on an anti-CD3 and anti-CD28 antibody–coated plate for one day and incubated with tenfold more HIV-1. Resting CD4⁺ T cells from the peripheral blood express HIV-1 infection-restricting SAMHD1, a dNTP triphosphohydrolase[92], requiring a much higher virus amount.

### Fluorescence-activated cell sorting (FACS) analysis

CD4⁺ T cells were purified from the buffy coat of healthy individuals. Percentages of CD4⁺ T cells were monitored before and after purification by staining with anti-TCRβ and anti-CD4 antibodies.

Cultured cells were harvested, washed with PBS + 1% fetal bovine serum (FBS) and with PBS + 0.1% glucose, and then resuspended in 200 μL of PBS + 0.1% glucose. After adding 5 mL of 70% ethanol (−20 °C) dropwise, samples were incubated at −20 °C for more than 30 min, washed twice with PBS + 0.1% glucose +1% FBS, resuspended in 0.5 mL of a propidium iodide/RNase staining buffer (Becton Dickinson), and incubated at 37 °C for 30 min. The resulting samples were filtered through a 50-μm nylon mesh and analyzed using the BD LSRFortessa flow cytometer (Becton Dickinson). Obtained data were analyzed by the FlowJo 10.9.0 software. All the FlowJo-analyzed data are provided in the Source Data.

### Immunostaining

Immunostaining was performed as described previously[36]. In brief, U2OS or TZM-bl cells were grown on poly-L-lysine (Sigma-Aldrich)-coated No. 1.5 coverslips, fixed with 4% paraformaldehyde, permeabilized with 0.1% Triton X-100 for 5 min, and then blocked with 5% bovine serum albumin in phosphate-buffered saline. The cells were stained with the indicated primary antibodies and appropriate Alexa fluorophore–conjugated secondary antibodies (Thermo Fischer Scientific) listed in Supplementary Table 3. The resulting samples were mounted with ProLong Gold Antifade (Thermo Fisher Scientific) before microscopic analysis.

To prepare thin-layered samples with suspension cultures (CEM-SS and Human CD4⁺ T cells), $0.25 \times 10^6$ cells (400 μL) were put into a cuvette assembled with a poly-L-lysine (Millipore Sigma)-coated No. 1.5 coverslip (Marienfeld Superior) and a cytoclip slide clip, then centrifuged at $300 \times g$ for 4 min using Cytospin 3 (Thermo Scientific Shandon). The resulting cells mounted on coverslips were fixed with 4% paraformaldehyde or 1:1 methanol and acetone mixture (for γ-tubulin staining only) and then immunostained as described above.

### Confocal microscopy and three-dimensional structured illumination microscopy (3D-SIM)

Confocal images were acquired under the confocal mode of the Zeiss ELYRA S1 super-resolution microscope (Zeiss) equipped with an Alpha Plan-Apo 63×/1.46 oil objective, 405 nm/488 nm/561 nm/ 640 nm laser illumination, and standard excitation and emission filter sets. To quantify fluorescence signal intensities, images were acquired under the same laser intensities, converted to the maximum intensity–projected images of multiple z-stacks, and then analyzed using the Zeiss Zen v2.1 software.

For 3D-SIM microscopy, images were acquired by the same ELYRA S1 microscope and then processed using the ZEN black software (Zeiss).

### TEM tomography and segmentation

**Sample preparation for electron microscopy.** U2OS cells stably expressing mGFP-Vpr grown on 35 mm gridded glass bottom dishes (MatTek Corporation) were subjected to light microscopy imaging. The resulting cells were fixed with Karnovsky's fixative, then post-fixed, stained, and resin-embedded as described previously[93], except that 2% OsO₄ and 1% uranyl acetate were used. In addition, a graded series of Polybed resin (EMS) with ethanol was utilized (2:1 ethanol to resin, 1:1 ethanol to resin, and 1:2 ethanol to resin each for 1 h). Finally, the cells were incubated at room temperature for approximately 48 h in resin without activator, followed by a 4-hour incubation in resin with activator BDMA at 32 °C. The cells had a final exchange with degassed resin and were allowed to polymerize for 24 h at 65 °C. The resin was separated from the glass coverslip by heat shock, and any remaining glass particles were removed by hydrofluoric acid treatment and subsequent washing in ddH₂O. Regions of interest identified by light microscopy imaging were cut out with a jeweler's saw and glued to resin blocks using super glue. Serial sections were cut *en face* on an ultramicrotome and collected on formvar-coated slot grids. Grids were post-stained with uranyl acetate and lead citrate according to standard protocols, then were carbon coated before being transferred to the TEM for tomography.

**Tomographic collection and reconstruction.** Areas identified as targets were first exposed to the microscope beam for 10 min before tilt series collection to induce resin section collapse (baking) before imaging. Tilt series were collected on a 120 kV Talos L120C TEM (Thermo Fisher Scientific). Images were recorded using a Ceta 16 M CCD camera. Tilt series were recorded using SerialEM[94] software at a magnification corresponding to a pixel size of 0.966 or 0.595 nm with 1-degree tilt increments over an angular range of ±60° by using a Model 2020 advanced tomography holder (Fischione). Tilt series were reconstructed by using Etomo in the IMOD software package, version 4.11[95], utilizing patch tracking and weighted back-projection with the simultaneous iterative reconstruction-like filter for the final 3D volume. Serial reconstructed volumes were joined using the join serial tomograms module in Etomo. Tomogram movies were created using Quicktime.

**Segmentation of centrioles.** Before segmentation, tomograms were binned volumetrically by 2 or 4 in IMOD and trimmed to the sub-volume containing centriole clusters. The processed tomograms were loaded into 3D Slicer Version 5.0.2 (https://www.slicer.org)[96], and median and Gaussian blur filters were applied. The centrioles and matrix around the centrioles were segmented by using threshold-assisted painting. The resulting 3D models were used to create figures and movies.

### Metaphase chromosome spread

Samples for metaphase chromosome spread were prepared as described previously[97]. In brief, CD4⁺ T cells infected with HIV-1 WT or HIV Vpr(1–79) were cultured by plating the cells on an anti-CD3 and anti-CD28 antibody–coated plate for 4 days. Uninfected control cells were simultaneously prepared by following the same procedure without the virus. The resulting cells were treated with 0.1 μg/mL of colcemid (Thermo Fisher Scientific) for 45 min, harvested, then resuspended in a prewarmed 75 mM KCl (hypotonic solution) at 37 °C for 8 min. The samples were washed three times with freshly prepared fixative (3 methanol:1 glacial acetic acid), mounted on a slide in a drop-by-drop manner, and stained with Giemsa stain solution (Thermo Fisher Scientific). Images were acquired using the Keyence inverted fluorescence phase contrast microscope BZ-X710 equipped with a 20× objective lens (zoom 3×).

## Immunoprecipitation and immunoblotting

Immunoprecipitation was carried out as described previously[98] in TBSN buffer (20 mM Tris-HCl [pH 8.0], 120 mM NaCl, 0.5% Nonidet P-40, 5 mM EGTA, 1.5 mM EDTA, 2 mM DTT, 20 mM *p*-nitrophenyl phosphate, and protease inhibitor cocktail [Roche]). Immunoblotting was performed using enhanced chemiluminescence reagents (Thermo Fisher Scientific), and signals were captured using a chemiluminescence imager (ChemiDoc™ Imaging Systems, Bio-Rad Laboratories). Where indicated, the signal intensities in the immunoblots were quantified using the ImageJ (NIH) or Image Lab (Bio-Rad) 6.0.1 software. All the antibodies used in this study are listed in Supplementary Table 3.

## ZZ/FLAG-affinity purification and mass spectrometry

For ZZ-tag affinity purification of Plk4 CPB (581–884) (pKM4285) or its control vector (pKM4283), total cellular lysates prepared from HEK293T cells transfected with the respective construct were subjected to affinity purification with a human IgG (Amersham Biosciences) column. After digestion with AcTEV protease (Invitrogen), proteins were eluted with TBSN buffer for further analysis. For FLAG-affinity purification of Plk4 CTD (581–970) (pKM4591) or its control vector (pKM4067), HEK293T cells that stably expressed them using a lentiviral expression system were subjected to α-FLAG immunoprecipitation. Affinity-purified proteins were separated by 10% sodium dodecyl sulfate-polyacrylamide gel electrophoresis (SDS-PAGE). The protein bands of interest excised from the gel were subjected to in-gel digestion with trypsin (Promega, Madison, WI) followed by nanoLC-tandem mass spectrometry, as described previously[85]. All the identified peptides are in MassIVE (accession number MSV000094032), and some are shown in Fig. 2a and Supplementary Fig. 2a.

## Protein expression and purification

Purification of CPB (581–808) using the His-MBP-TEV-CPB(581–808) construct (pKM3677) was described previously[37]. To purify His-CPB (pKM5444), His-MBP-CPB (pKM3686), His-MBP-Plk4 WT (pKM4601), and His-MBP-Plk4 K41M (pKM4602), *E. coli* Rosetta strains (Novagen) expressing the respective constructs were cultured and the proteins were induced with 1 mM Isopropyl β-D-1-thiogalactopyranoside overnight at 16 °C. The resulting cells were lysed in an ice-cold buffer (20 mM Tris-HCl [pH 7.5], 700 mM NaCl, 0.5 mM TCEP), and the lysates were subjected to HisTrap HP column (GE Healthcare) and HiLoad 16/60 Superdex 200 (GE Healthcare) SEC. Purified proteins were stored in the final buffer (20 mM Tris-HCl [pH 7.5], 700 mM NaCl, 0.5 mM TCEP) at −80 °C until use.

VprBP and Vpr were expressed in an improved insect cell line, *Trichoplusia ni* (Tni)-FNL[99]. To purify the protein, transfected insect cells were cultured in Sf-900 III SFM at 27 °C for 2.5 days, lysed in an ice-cold buffer (20 mM Tris-HCl [pH 7.5], 500 mM NaCl, and 10% [v/v] glycerol), and then subjected to the HisTrap HP column (GE Healthcare). The HisTrap eluate was digested via TEV, mixed with Plk4 CPB, and then subjected to SEC in 20 mM Tris-HCl (pH 7.5), 100 mM NaCl, and 10% (v/v) glycerol.

## Ubiquitination assays

The CRL4-VprBP-Vpr E3 ligase was assembled by mixing equal molar amounts of separately purified CUL4A-RBX1, DDB1-VprBP (full-length), and Vpr as reported previously[49,100,101]. E1 (UBA1) and E2 (UBC5B) were purified as previously described[49]. Typically, E1 (0.8 μM), E2 (5 μM), and appropriate E3 ubiquitin ligase (0.6 μM) were incubated with 2 μM Plk4 or UNG2 and 5 μM ubiquitin in a reaction buffer containing 10 mM Tris-HCl (pH 7.5), 150 mM NaCl, 5% glycerol, 20 units/mL pyrophosphatase, 1 mM Tris(2-carboxyethyl)phosphine, and 5 mM ATP. The reactions were terminated with SDS-PAGE sample loading buffer, and immunoblotting with appropriate antibodies revealed the extent of ubiquitination.

## In vitro kinase assay

Kinase reaction was carried out essentially as previously described[42] with Sf9-purified GST-Plk4ΔPB3 (residues 1–836) (Sigma-Aldrich) in vitro in a kinase cocktail containing 50 mM Tris-Cl (pH 7.5), 10 mM MgCl₂, 2 mM EGTA, 2 mM DTT, and 100 μM ATP. For the time-course kinase reactions in Supplementary Fig. 5a, left, reactions were performed with 54 nM GST-Plk4ΔPB3 at 30 °C and terminated by the addition of 5× Laemmli SDS sample buffer. The reaction products were separated by 4−15% SDS-PAGE for silver staining and immunoblotting analyses. Unphosphorylated Plk4 was prepared by reacting with λ phosphatase (λ).

For kinase reaction in Supplemental Fig. 5a, right, reactions were conducted in the presence of Sf9 cell-expressed Vpr•VprBP or Vpr(1−79)•VprBP at the indicated molar concentrations (1:1 or 20:1) with 54 nM GST-Plk4ΔPB3. Reacted samples were separated by 4−15% SDS-PAGE and subjected to silver staining and immunoblotting analysis to determine the level of the p-SSTT epitope generated during the reaction. The p-SSTT signals were quantified using the Image Lab 6.0.1 software (BioRad).

## iSCAMS

The iSCAMS measurement was carried out using a Refeyn TwoMP mass photometer (Refeyn Ltd.). Cleaned coverslips were assembled into flow chambers. All the buffers used for analysis were filtered through a syringe filter with 0.45-μm pore size (Anotop 10, Whatman). For measurements, all samples were freshly diluted from stock solutions. Sample proteins (100 nM) were diluted to the final concentration of 20 nM in 20 mM HEPES (pH 7.5) buffer containing 200 mM or 400 mM NaCl. The protein solution was loaded into the sample well after finding the focus with the buffer. Obtained data were processed with the DiscoverMP program (Refeyn Ltd.)

## Statistics and reproducibility

All the experiments were performed at least three times independently. All values are given as mean of $n \pm$ s.d. $P$ values were calculated by unpaired two-tailed *t*-test from the mean data of each group. All the co-immunoprecipitation and binding analyses shown in Figs. 2b–e, 3a–e, and 6c were repeated more than three times. Representative blots and gels are provided.

## Reporting summary

Further information on research design is available in the Nature Portfolio Reporting Summary linked to this article.

## Data availability

Data supporting the conclusion of analyzing the samples obtained from people living with HIV-1 are provided in Supplementary Fig. 1. To protect study participants' personal information, only three indirect identifiers are provided. All the raw mass spectrometry data used to generate Fig. 2a and Supplementary Fig. 2a have been deposited in MassIVE with accession number MSV000094032 (https://massive.ucsd.edu/ProteoSAFe/dataset.jsp?task=9e6e062307f94ab1be7f2a7db40f257e). All the raw data used for quantification and statistical analyses are provided in the Source Data file. Source data are provided with this paper.

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

## Acknowledgements

We are grateful to Inwoo Park for sharing his expertise on HIV-1, Jadranka Loncarek and Mirit I. Aladjem for offering antibodies, Di Wu and Grzegorz Piszczek for assisting with iSCAMS analysis, and Michael Kruhlak and Langston Lim for helping with 3D-SIM and confocal image analyses. This research was supported by the Intramural Research Program of the National Institutes of Health (NIH) (K.S.L., K.S., J.D.A., and F.M.) and a federal fund from the National Cancer Institute, NIH, under Contract No. 75N91019D00024 (K.N.).

## Author contributions

J.-E.P., T.-S.K., Y.Z., C.M.M., J.I.A. and K.S.L. designed all the experiments. J.-E.P. performed all cell biology analyses, while T.-S.K. carried out all the coimmunoprecipitation analyses. Y.Z. and C.M.M. conducted iSCAMS-based analyses and in vitro ubiquitination assays, respectively. M.A. and V.S. purified and cultured primary CD4+ T cells, and M.Z. performed and analyzed mass spectrometry data. M.M. and K.N. performed TEM tomography and analyzed the data. F.M. and T.-W.C. organized PBMCs from people living with HIV-1, and J.D.A. offered his expertise in T-cell-related studies. K.S. shared his Vpr reagents and offered advice on various HIV-1-related experiments. F.M., J.D.A. and K.S. provided valuable insights during the course of this work. K.S.L., J.-E.P., T.-S.K., Y.Z., J.A., M.M. and K.N. wrote the manuscript.

## Funding

## Competing interests

The authors declare no competing interests.

## Additional information

¹Cancer Innovation Laboratory, Center for Cancer Research, National Cancer Institute, National Institutes of Health, Bethesda, MD 20892, USA. ²Center for Molecular Microscopy, Center for Cancer Research, National Cancer Institute, National Institutes of Health, Bethesda, MD 20892, USA. ³Cancer Research Technology Program, Frederick National Laboratory for Cancer Research, Frederick, MD 21702, USA. ⁴Laboratory of Immune Cell Biology, Center for Cancer Research, National Cancer Institute, National Institutes of Health, Bethesda, MD 20892, USA. ⁵Department of Structural Biology, University of Pittsburgh School of Medicine, Pittsburgh, PA 15260, USA. ⁶Laboratory of Immunoregulation, National Institute of Allergy and Infectious Diseases, National Institutes of Health, Bethesda, MD 20892, USA. ⁷Protein Characterization Laboratory, Frederick National Laboratory for Cancer Research, Frederick, MD 21702, USA. ⁸HIV Dynamics and Replication Program, National Cancer Institute, National Institutes of Health, Frederick, MD 21702, USA. ⁹Laboratory of Molecular Microbiology, National Institute of Allergy and Infectious Diseases, National Institutes of Health, Bethesda, MD, USA. ¹⁰These authors contributed equally: Jung-Eun Park, Tae-Sung Kim. ✉e-mail: kyunglee@mail.nih.gov

