## [Peer Review File · Nature Communications]

Centrosome amplification and aneuploidy driven by the HIV-1-induced Vpr•VprBP•Plk4 complex in CD4⁺ T cellsREVIEWER COMMENTS

Reviewer #1 (Remarks to the Author):

In this study, Park et al. (2023) identified VprBP as an interacting protein of Plk4/Plk4CPB. VprBP is known to bind to Vpr, an accessory small protein of HIV-1. The presence of modest centrosome amplification in the blood cells of HIV-infected patients suggests its potential role as a driver of T-cell lymphoma. The authors demonstrate that the ternary protein complex comprising Vpr, Vpr BP, and Plk4 regulates the Plk4 ring-to-dot conversion process, leading to centrosome amplification. Furthermore, the study indicates the indispensability of both Vpr and VprBP for centrosome amplification. Notably, the authors observed that HIV-1 WT, but not a Vpr mutant incapable of binding to Plk4, induces centrosome amplification in cells.

Overall, the biochemical data supporting the formation of the ternary complex exhibit high quality and convincingly validate its existence. However, the mechanisms through which the ternary complex regulates Plk4 function in centrosome amplification remain unclear. Moreover, the evidence establishing the altered Plk4 function as the primary cause of centrosome amplification in Vpr-transfected U2OS cells and HIV-infected primary T cells is insufficient. Consequently, the following concerns preclude publication in Nature Communications, and should be experimentally addressed.

Main points:

1. In Fig. 3d, there appears to be a significant reduction in the total expression level of Flag-Plk4 when expressing Flag-Vpr and GFP-VprBP in the cells. A similar trend is observed in the input of Fig. 3b, although the difference is less pronounced. Can the authors confirm the reproducibility of these findings? If so, is it possible that the formation of the ternary complex targets Plk4 for protein degradation, despite an enhancement in the complex formation itself?

2. It is intriguing that Vpr increases the amount of Flag-Plk4 in the IP fraction of GFP-VprBP by approximately 10-fold. How does a small protein like Vpr facilitate the interaction between these two proteins? In Fig. 3C, it appears that the full-length Vpr is necessary for efficient ternary complex formation, with the C-terminus of Vpr exhibiting significantly weaker ability compared to the full-length form. The model presented in Fig. 3f may not accurately represent the ternary complex, as Vpr might alter the interaction mode between VprBP and Plk4 in a more complex manner, possibly by binding to WD40 of VprBP. Have the authors conducted direct in vitro tests to examine the interaction between Plk4 and Vpr?

3. It would be informative if the authors could quantify the kinase activity of Plk4 in the ternary complex (versus Plk4 alone) both in vitro and in cells.

4. In Fig. 4d, the addition of MG132 is mentioned, but the reason for its inclusion is not provided. Please clarify the rationale behind the use of MG132. Additionally, it is essential to ascertain whether the cell cycles under each condition are comparable. Since ring-like localization of Plk4 is observed in early G1 phase, any alterations in the cell cycle duration resulting from the expression of Vpr could influence the analysis appropriately. Therefore, the authors should investigate whether the expression of mEGFP-Vpr affects the duration of G1 phase. Alternatively, the localization of Plk4 within the same cell cycle stage could be analyzed.

5. In the manuscript, the authors claim that Vpr and VprBP promote the dot-like localization of Plk4. However, the connection between the dot-like localization of Plk4 and centriolar overduplication is unclear. It has been hypothesized that the dot-like distribution of Plk4 restricts the potential sites of centriole overduplication and may even suppress centriolar overduplication. Therefore, comprehending their model is not straightforward. The authors should explain the potential mechanisms through which changes in Plk4 distribution lead to centriole overduplication.

6. Previous studies have reported the interaction between Vpr and Cep78 mediated by the binding of Vpr to VprBP (Hossain et al., 2018, J Biol Chem). It has also been shown that the expression of

a Vpr mutant (Q65R) lacking interaction with VprBP and Cep78 fails to induce centriole overduplication (Hossain et al., 2018, J Biol Chem). Additionally, Cep78 has been implicated in Plk4-mediated centriole overduplication (Brunk et al., 2016, J Cell Sci). Thus, it appears that Cep78 plays a significant role in HIV-mediated centriole overduplication. To fully understand the mechanisms underlying HIV-induced centriole overduplication, the authors should consider the involvement of Cep78 and conduct experimental investigations to address this aspect.

7. In Figure 4, the authors demonstrated that Vpr(1-79), which lacks interaction with VprBP and Plk4, does not induce centriole amplification. This finding suggests that the interaction between Vpr and Plk4 through VprBP promotes centriole overduplication. However, considering that the C-terminus of Vpr and VprBP may interact with other proteins, including Cep78 (Hossain et al., 2017, EMBO Rep), it is not appropriate to conclude that the Vpr-VprBP-Plk4 complex solely drives centriole overduplication. If feasible, it would be helpful to examine a Vpr variant with a point mutation that lacks interaction with Plk4 but can still interact with Cep78.

Minor points:

1. In Figure 4b, the authors did not include the quantification of centrosomal Plk4 intensity in cells expressing only mEGFP (without Vpr) as a control. It is important to determine whether the expression of mEGFP-Vpr affects the centrosomal intensity of Plk4 compared to the expression of mEGFP alone. Including this control would help assess the specific impact of mEGFP-Vpr on Plk4 localization.

2. In Fig.4b, a difference in the Plk4 intensity between control and shVprBP does not seem to be significant (n=3 or more than 3), compared to that in the Vpr intensity. It would be helpful if the authors could provide an explanation for this observation.

3. Regarding the EM images shown in Figure 5a, centrioles 1 and 2 appear to be mature or elongated, while the other amplified centrioles seem to be shorter or immature. It would be beneficial if the authors could provide an interpretation of these observations. Do these findings reflect the nature of the centrosome amplification phenotype induced by GFP-Vpr expression? Additionally, it would be informative to know how the amplified centrioles in HIV-infected or patient cells compare to the ones observed in this study.

4. In cells expressing GFP-Vpr, do the amplified centrioles have the ability to induce multipolar spindle formation during mitosis? Since centriole amplification can potentially lead to the formation of extra spindle poles, it would be relevant to investigate whether the amplified centrioles in GFP-Vpr-expressing cells contribute to multipolar spindle formation during cell division.

5. The authors could examine whether the distribution of Plk4 around centrioles is affected in HIV-infected or patient cells. Investigating the impact of HIV infection on the localization and distribution of Plk4 around centrioles would provide valuable insights into the cellular mechanisms underlying centrosome amplification in the context of HIV infection or T-cell lymphoma in patients.

Reviewer #2 (Remarks to the Author):

In this manuscript by Park and Kim et al., the authors find that a small proportion of CD4+ T cells isolated from HIV-1 infected individuals display centrosome amplification. Interestingly, this phenomenon is absent in healthy control cells. Given the known association of Plk4 and centrosome duplication, the authors perform biochemical analyses to identify and characterise a ternary complex formed by Vpr-VprBP-Plk4. They also show that Vpr can induce multi-centromeres in model cell lines and primary CD4+ T cells.

This is an interesting and important study and, overall, well performed. However, it could be strengthened to convince the reader of some of the phenotypes.

Figure 1 sets the scene for the paper, and some of the data from this figure are hard to interpret. The age of the HIV-1-infected individuals, their additional co-morbidities, and all details from the healthy control samples analysed were absent. It would be helpful to know how accurately the healthy controls matched the infected individuals. Additionally, the authors draw their conclusions after analysing two healthy controls.

Given the low number of cells analysed (around 300 for HIV-1-positive individuals) and the modest percentage of cells with multi-centrosomes (around 1%) for 3/5 samples, the authors detected 3-6 cells with multi-centrosomes. Therefore, the power of these observations (combined with the lack of information about some of the samples) seems weak. These data are essential for the story's development, so it is important for the authors to strengthen these data by including additional healthy controls (clearly stating how they were matched to HIV-1-positive individuals). It would also be helpful to have additional samples from HIV-1-positive individuals at different stages of the disease. Control samples should be relatively easy to obtain, so a 3:1 ratio control:case would provide more confidence in the data.

Importantly, the majority of lymphomas in patients with HIV are high-grade B cell lymphomas, plus (a minority of) Hodgkin's i.e. they do NOT arise from HIV-infected CD4+ T cells (some are related to herpesviruses, particularly EBV and HHV-8). In this paper most (possibly all) of the features they describe relate to cell-autonomous effects i.e. in principle, relating to HIV-infected CD4+ T cells. They address this issue here:

Therefore, showing that this might/could occur in uninfected cells would seem essential e.g. (as a minimum) Vpr-VLPs in T cells, or of more relevance B cells (this would have to be with "free" Vpr). Otherwise it's unclear how their interesting observation relates to the issue they wish to address – how HIV-infected individuals are more susceptible to a range of, predominantly B-cell derived lymphomas?

Are the abnormalities they describe actually seen in and/or characteristic of HIV-associated lymphomas (or solid cancers) – I couldn't find this information? They say they look at cells from 5 HIV positive donors, although details about these patients seemed to be absent e.g. what stage of infection, were they viraemic, on treatment, have advanced HIV, etc.

Other points:

The biochemical data presented in Figures 2 and 3 is convincing and well-performed.

The microscopy data in Figures 4, 5 and, 6 is less convincing.

In some figures, it is unclear which phenotype the authors are showing. For example, in Supplementary Figure 5A, the authors mention: "Vpr robustly induced centrosome overduplication and VprBP further augmented it (Fig. 5b and Supplementary Fig. 5a)". However, I find it hard to see that in the images shown, and am concerned about the quantification.

For all this section, it would help to include a CellTracker staining such as the one in Inanç et al., MBOC 2010. (In some way, in Supplementary Figure 6, p24 staining acts as a CellTracker and greatly facilitates the interpretation of the images).

In figures showing mGFP-Vpr (e.g. 4D, 4E), adding lower magnifications, DIC, and DAPI is important. It is hard to understand which part of the cell is being shown. The difference in localisation between mGFP-empty and mGFP-Vpr (if any) is hard to see.

The authors state: "Notably, mGFP-Vpr alone effectively localized to the pericentriolar material (PCM) region, often showing a "nebulous" appearance that encompasses multiple dot-like Plk4

signals (Fig. 4d and Supplementary Fig. 4f). In contrast, the Vpr(1-79) mutant localized poorly to centrosomes and failed to significantly induce the dot-like Plk4 signals (Fig. 4d and Supplementary Fig. 4f).". However, this observation is not obvious when looking at the images displayed. An alternative interpretation of the images would be that they both poorly localise at the centrosomes or that Vpr(1-79) better localises at the centrosomes. Do authors have better representative images to show this phenotype?

Why is the distribution of mCh-VprBP more cytosolic when co-expressed with mGFP-Vpr(1-79)? Is the change in the distribution of mGFP-Vpr when co-expressed with mCh-VprBP(1-1427) representative?

For Figure 5a, controls and quantification were not included. Additionally, the authors state: "The result showed that some cells exhibit up to 7 centrioles". However, only one cell was analysed (according to the data shown). As is, these data would be better fit as supplemental information in Figure 4.

Again, for Figure 6C, adding lower magnification, DIC, and DAPI is critical. Are the authors showing only a single cell?

Is Plk4 expressed in primary CD4+ T cells? Can the authors include this antibody in the western shown in Supplemental Figure 6D? (also, a small suggestion would be to rename "PBMCs CD4+ T cells" to "primary CD4+ T cells, to avoid confusion).

I couldn't find the information about the HIV-1 plasmids used in Figure 6. How did the authors reconstitute the coding sequence of Tat in the Vpr(1-79) mutant (overlapping orfs)? What percentage of the primary CD4+ T cells are infected for each condition? Are the cells shown in Figure 6C infected? Can the authors show they are infected? Is this data representative of multiple donors? For Figure 6D, were the cells first selected based on infection status?

Many of these questions are aimed at understanding if the authors believe this is a cell-autonomous or non-cell-autonomous effect. Vpr is packed in virions at equimolar ratios with Gag. Can Virus-like Particles (VLPs) packed with Vpr reproduce the effects seen?

Could they make a longitudinal analysis for some of the samples analysed?

In the discussion, the authors say: "In addition, since a substantial level of Vpr (10 pg/mL-10 ng/mL of blood) is present in the blood of people living with HIV-1", 68-70. It could be important to mention that Vpr levels correlate with RNA levels 69 and that ARV should bring both factors to undetectable levels.

Reviewer #3 (Remarks to the Author):

In this study, Park and colleagues investigate the mechanisms underlying previous findings indicating that HIV-1 infection and notably the virus-encoded Vpr accessory protein, disrupts centrosome homeostasis, a condition that may contribute to oncogenesis by driving aneuploidy. They provide evidence that CD4+ T lymphocyte isolated from the blood of persons living with HIV (PLWH) display to varying extent over-duplicated centrioles although as discussed below it is difficult to assess the specificity of this effect as these observations were made on the total CD4+ T cell population (HIV-positive and HIV negative) and it is unclear if the two healthy controls were age-matched with the five PLWH. Consistent with previous studies, they show that HIV-1 Vpr induces centrosome amplification; importantly, they uncover a novel mechanism used by Vpr to hijack the centriole duplication machinery and induce centrosome amplification and aneuploidy. Using biochemical and cellular biology analyses, they provide novel and strong evidence that Vpr

forms a cooperative ternary complex with VprBP, the substrate recognition receptor of the CUL4-DDB1 E3 ubiquitin ligase complex, and the Polo-kinase-like 4 (Plk4), a key regulator of centriole biogenesis that is found to form a complex with VprBP in normal conditions. Indeed, the authors provide evidence that the formation of a ternary complex with Vpr does not lead to Plk4 ubiquitination and degradation but rather enhance Plk4 functionality by driving its relocation to the procentriole assembly site. By defining and characterizing the domains of Vpr, VprBP and Plk4 that are involved in the formation of this ternary complex they further demonstrate that the loss of complex formation abrogates Vpr ability to induce centriole over-duplication. To underline the biological significance of their findings, they show that WT HIV but not Vpr mutants that are unable to form a ternary complex with VprBP and Plk4, induce centrosome amplification and aneuploidy in a fraction of infected cells in various cellular model of HIV infection, including primary CD4+ T cells.

Overall, this is an interesting, and well executed study that uncover a novel host target of Vpr (Plk4) and a novel mechanism by which this viral encoded accessory protein hijacks the centriole duplication machinery, inducing centriole over-duplication and aneuploidy, independently of its ability to induce a cellular growth arrest at the G2 stage of the cell cycle. While the biochemical characterization of the Vpr-VprBP-Plk4 complex and the impact of various mutants on Vpr capacity to disrupt centrosome biogenesis represent strong components of the study, the extrapolation of these observations obtained in model systems in vivo remains rather weak. Additionally, while aneuploidy can be a driver of oncogenesis, the authors do not address the apparent paradox of Vpr mediating such a process but inducing at the same time a cellular growth arrest and apoptosis in HIV-infected cells.

Specific points:

1-Figure 1 and supplementary figure 1. Authors detect more duplicated centrioles (1-5%) within the total CD4+T cells from people living with HIV when compared with normal healthy donor. It is unclear from the information provided if the healthy donor were age-matched as age could influence the comparison. It seems also that the frequency of centriole duplication is not linked to viral load. as observed in patient 3. Was the number of years living with HIV considered in the analysis? In the absence of evidence that the centriole over-duplication is occurring primarily in HIV+ CD4+ T cells it is difficult to assess whether these observations are specific to HIV infection or other confounding effects.

2-Figure 2: panel 2b Except for the NTD Flag- PLK4, and a very weak signal for CPB Flag-Plk4, no other Plk4 mutants are detected in the input fraction. Is there a difference in transfection efficiency between the different constructs? Moreover, the IP for the full length Plk4, reveals multiple bands below 116kDa. Are there multiple Plk4 isoforms that could interact with VprBP?

3-Supplementary Figure 3b: this figure is aimed at demonstrating that Vpr-enhancement of Plk4 association to VprBP is not dependent on Plk4 catalytic site. Why perform a Plk4 pull down and not a VprBP pull down as shown in figure 3b to clearly show the enhanced association of Plk4 to the complex in the presence of Vpr. As it is the enhanced recruitment of Plk4 WT and mutant is not obvious.

4-Figure 6A: although there is a significant increase in the number of centrosomes in HIV-1 Vpr WT infected cells, there is also an increase in the number of cells harboring 3 to 4 centrosomes in Vpr (1-79) and Vpr (-) infected cells that seems significant when compared to uninfected cells. Could the authors explain why in absence of Vpr but during HIV infection, there seems to be a deregulation in centrosome number?

5-Figure 6: the authors show the p24 levels by western blot in the supplementary figure 6d and state that approximately 10% of the CD4+ T cells were infected. Figure 6C reveals that approximately 20% of p24+ cells display centrosome amplification, suggesting that about 2% of infected primary CD4+ T cells exhibit multi-centrosomes. However, analysis of chromosome numbers in HIV WT-infected primary CD4+ T cells indicate a reduction of 20% of total CD4+ T cells carrying a normal number of chromosomes and a concomitant increase in the frequency of cells carrying 43, 44, 45 chromosomes (less than 10% for each). Based on these numbers, it seems that evidence of aneuploidy is detected beyond infected cells as well as cells showing

centrosome amplification. Could the authors comment?

Minor Points

- 1- Figure 3d: the Vpx variant used should be specified in the figure legend
- 2- Discussion : The authors refer to several studies suggesting that PLWH are at greatly increased risk of developing in T cell lymphomas. Does the prevalence of these HIV-associated cancers decrease upon antiretroviral therapy? This would be important to discuss in light of the author hypothesis that HIV-encoded factors might be involved.

Reviewer #4 (Remarks to the Author):

The authors found that the HIV-1 accessory protein Vpr hijacks the centriole duplication mechanism, causing centrosome amplification and aneuploidy. The author claimed that Vpr could form a cooperative ternary complex with an E3 ligase component, VprBP, and polo-like kinase 4 (Plk4) and that the complex improved Plk4 functioning by encouraging relocalization to the procentriole assembly and inducing centrosome amplification. This is an intriguing concept; however, other aspects of the document contradict previous Vpr literature and are difficult to explain. Overall, there are significant flaws that, in my view, render this research inappropriate for Nat Comm at this stage. Specific remarks are provided below.

Major

1. The Fig. 1 is unconvincing. To infer that HIV-1 generates over-duplicated centrosomes in CD4+T cells in untreated HIV-1-infected persons, they must assess the levels of centrosome over-duplication in CD4+T cells from ART-treated patients since their viruses are inactive. According to the authors' premise, they should not find over-duplicated centrosomes in CD4+T cells from ART patients.
2. The figures in this paper (Figs. 2, 3, 4, and 5) are not physiologically relevant to HIV-1. At the very least, the authors should repeat certain crucial T-cell tests. Show how Vpr-VprBP-Plk4 interact in CD4+T cells. It is a crucial experiment for us to assume that this has physiological significance. Finally, the authors proved in Fig. 6 that HIV-1 infection increases centrosome duplication. Unfortunately, this does not persuade. They should utilize HIV-1 infection, whether EFV is present or not. Please do not use HIV-1 that has not been infected as a control. Please confine HIV-1 infection to a single cycle as well.
3. Why did the author investigate HIV-2 Vpx? This is quite unusual! Vpr is also found in HIV-2 and SIV! For this assumption, please erase the Vpx data and compare HIV-2 and SIV Vpr to HIV-1 Vpr. Furthermore, they should employ Vpr proteins from multiple HIV-1 strains (NL4-3, AD8, 89.6) to back up their findings.
4. Please investigate whether or not the virion-associated Vpr generates this behavior. There is no generated Vpr with raltegravir during HIV-1 infection, just encapsidated Vpr. Vpr is encapsulated in HIV particles, and the author must determine which Vpr induces centrosome amplification.
5. Vpr does not seem to have a profile that promotes HIV-1 infection in CD4+T cells (PMID: 29669271), but it does increase HIV-1 infection in primary macrophages. The authors should additionally investigate if Vpr binds to PLK4 in non-proliferating macrophages (no centrosome?). This might be one reason for Vpr's different involvement in CD4+T and macrophages.
6. Fig. 4. Plk4 kinase has specific inhibitors! Please use these inhibitors to see whether kinase activity is required for HIV-1-induced over-duplication of centrosomes. This is critical.
7. Please provide MOI for HIV-1 infection; it is critical to determine whether the phenotype is physiological or not.

Minor:

1. From Fig. 1, we can observe that there were no multi-centrosomes in the healthy case group. However, it is unclear if the cells in this image were infected with HIV-1, despite the fact that not all cells in HIV patients were infected with the virus. More evidence is required.
2. Based on Fig. 1, the author hypothesized that Plk4 was unregulated during HIV-1 infection. Did they examine other plausible mechanisms, such as "accidental" activation of CDK2/cyclin E, which causes centrosome re-duplication and, perhaps, centrosome amplification? Is it feasible that additional PLK family proteins are involved? In essence, the centrioles are "primed" for duplication in the early G1 phase by pericentrin cleavage, which leads to centriole disengagement. The cyclin-dependent kinase 1 (CDK1), polo-like kinase 1 (PLK-1) and Aurora kinase A control this process.
3. The author performed extensive IP experiments to illustrate the connection between PLK4 and VprBP in Fig. 2. Cell lines and exogenous transfection were used in all of these studies. There is no evidence for an interaction between endogenous PLK4 and VprBP, particularly in HIV-1-vulnerable target cells.
4. Similar questions were also included in Fig. 3. What is the endogenous level of the PLK4-VprBP-Vpr complex, particularly in HIV-1-vulnerable target cells? Is their interaction at HIV-1 Vpr+/- compatible with that seen in the cell line?
5. The author employed the HIV-2 Vpx protein as a negative control for VprBP binding in Fig. 3d, but they neglected one essential point. Vpr is widely known to be present in both HIV-2 and SIV. However, the author did not investigate whether HIV-2 and SIV Vpr could bind to VprBP. They did not even say if Vpr from various species and genera may trigger centrosome amplification.
6. The author only investigated the impact of PLK4 in Fig. 4, but this does not completely describe whether Vpr of other proteins in the PLK family have the same effect on it.
7. The Figs. 4d and 4e are likewise vulnerable. Will HIV-2/SIV Vpr trigger centrosome amplification as well? Even the auxiliary protein Vpx, which does not bind to VprBP?
8. Although the author detected centrosome amplification in activated CD4+T cells and U2OS cell lines, further controls are needed to clarify the critical function of Vpr, particularly in HIV-2/SIV.
9. It is remarkable that PLK4 is expressed poorly in Supplementary Figs. 6b and 6c, despite the fact that it plays a critical role in centrosome amplification, as the author of the research pointed out. Is this due to the action of Vpr on other host proteins, which causes centrosome amplification?
10. The authors stated that VprBP and PLK4 were both required for Vpr to generate numerous centrosomes. However, VprBP expression was substantially higher than PLK4, and the shift in PLK4 expression following Vpr was not visible in Supplementary Fig. 4. VprBP experienced the same thing. Endogenous expression of VprBP and PLK4 must be reevaluated.
11. Supplementary Fig. 6d: The immunoblots must contain a PLK4 (anti-PLK4) blot to ensure that all proteins are expressed at similar levels.
12. Please also investigate if Plk1 binds to Vpr since Plk1 is a member of the PLKs. Perhaps a nice Plk4 control.

REVIEWER COMMENTS

Reviewer #1 (Remarks to the Author):

In this study, Park et al. (2023) identified VprBP as an interacting protein of Plk4/Plk4CPB. VprBP is known to bind to Vpr, an accessory small protein of HIV-1. The presence of modest centrosome amplification in the blood cells of HIV-infected patients suggests its potential role as a driver of T-cell lymphoma. The authors demonstrate that the ternary protein complex comprising Vpr, Vpr BP, and Plk4 regulates the Plk4 ring-to-dot conversion process, leading to centrosome amplification. Furthermore, the study indicates the indispensability of both Vpr and VprBP for centrosome amplification. Notably, the authors observed that HIV-1 WT, but not a Vpr mutant incapable of binding to Plk4, induces centrosome amplification in cells. Overall, the biochemical data supporting the formation of the ternary complex exhibit high quality and convincingly validate its existence. However, the mechanisms through which the ternary complex regulates Plk4 function in centrosome amplification remain unclear. Moreover, the evidence establishing the altered Plk4 function as the primary cause of centrosome amplification in Vpr-transfected U2OS cells and HIV-infected primary T cells is insufficient. Consequently, the following concerns preclude publication in Nature Communications, and should be experimentally addressed.

Main points:

1. In Fig. 3d, there appears to be a significant reduction in the total expression level of Flag-Plk4 when expressing Flag-Vpr and GFP-VprBP in the cells. A similar trend is observed in the input of Fig. 3b, although the difference is less pronounced. Can the authors confirm the reproducibility of these findings? If so, is it

possible that the formation of the ternary complex targets Plk4 for protein degradation, despite an enhancement in the complex formation itself?

Response: Since multiple studies show that VprBP and Vpr function as a part of E3 ubiquitin ligases, we rigorously analyzed whether they could induce Plk4 degradation. Plk4 is a unique kinase whose *trans*-autophosphorylation can not only induce its self-destruction through β TrCP-mediated proteasomal degradation (Holland AJ, et al., 2010, JCB; Guderian G, et al., 2010, JCS) but also trigger its own liquid-liquid phase separation to form a condensate and induce centriole duplication (Park JE, et al., 2019, Nat Commun; Yamamoto S and D Kitagawa; 2021, Curr Opin Str Biol). While these two-faced biochemical events appear crucial for ensuring the binary on-off mode of Plk4-dependent centriole biogenesis, thus canceling out under-threshold Plk4 activities, they may render partially phosphorylated Plk4 susceptible to β TrCP-mediated degradation during its activation process (Guderian G, et al., 2010, J Cell Sci; Arquint C and Nigg EA, 2016, Biochem Soc Trans; Park JE, et al., 2020, Cell Cycle). To confirm the data provided in the original manuscript, we examined a Vpr expression level-dependent formation of a ternary complex with VprBP and Plk4. The new data provided in Fig. 3c and Supplementary Fig. 3b show that without substantially altering the amount of Plk4, Vpr interacted with Plk4 and VprBP. Furthermore, the level of the interactions among the three proteins was proportional to that of Vpr expression. The slight reduction in the level of Plk4 could be attributable to β TrCP-mediated Plk4 degradation, which occurs constantly in the cell.

Several lines of evidence we provided in the manuscript suggest that Plk4 is not the target of VprBP-mediated E3 ligase complexes. First, overexpression of Vpr and/or VprBP did not alter Plk4 stability (Supplementary Fig. 3j,k). Second, depletion of β TrCP, but not VprBP, increased the level of Plk4 in the total lysates (Supplementary Fig. 3l). Third, CRL4•VprBP•Vpr failed to increase the level of ubiquitinated Plk4 under the condition where it increased UNG2 ubiquitination (Supplementary Fig. 3m). Fourth, Vpr•VprBP, but not the Plk4 binding-defective Vpr(1–79)•VprBP, activated the catalytic activity of Plk4 *in vitro* (Fig. 4g and Supplementary Fig. 4p). Fifth, depletion of DDB1 or CUL4A, a component of the CRL4•VprBP•Vpr complex, failed to diminish Vpr-dependent centrosome overduplication (Fig. 5c and Supplementary Fig. 5c). These data are concisely stated on page 9 (“This new function of Vpr----”).

2. It is intriguing that Vpr increases the amount of Flag-Plk4 in the IP fraction of GFP-VprBP by approximately 10-fold. How does a small protein like Vpr facilitate the interaction between these two proteins? In Fig. 3C, it appears that the full-length Vpr is necessary for efficient ternary complex formation, with the C-terminus of Vpr exhibiting significantly weaker ability compared to the full-length form. The model presented in Fig. 3f may not accurately represent the ternary complex, as Vpr might alter the interaction mode between VprBP and Plk4 in a more complex manner, possibly by binding to WD40 of VprBP. Have the authors conducted direct in vitro tests to examine the interaction between Plk4 and Vpr?

Response: As the reviewer pointed out, the C-terminus of Vpr (Vpr CT) alone interacts with Plk4 weakly compared to the full-length Vpr. Although other possibilities, such as structural change upon forming the ternary Vpr•VprBP•Plk4 complex, could be possible, it is reasonable to speculate that the three-way interactions involving the full-length Vpr, but not the VprBP binding-defective Vpr(51–96) or Vpr(75–96), may have exhibited cooperative binding.

As suggested, we performed additional *in vitro* binding analyses with multiple Plk4 C-terminal fragments. The results provided in Supplementary Fig. 3i further support our original view. This new finding and other interaction data provided in Figs. 2 and 3 are consistent with the model shown in Fig. 6f. In particular, the interferometric scattering mass spectrometry (iSCAMS) data shown in Fig. 3f clearly demonstrate that the Vpr•VprBP•Plk4 complex forms with 1:1:2 stoichiometry within a minute after mixing the proteins and the formation of this complex is reversible (see also Fig. 3e).

However, it should be noted that Vpr is known to form homo-oligomers *in vivo* through the hydrophobic core formed by the three alpha helices (Fritz JV, et al., 2008, Retrovirology). This suggests that an additional layer could regulate the formation of the Vpr•VprBP•Plk4 complex, but most likely at a

concentration significantly higher than the concentration (100 nM) we used for iSCAMS analysis. This could be an area that needs further investigation.

3. It would be informative if the authors could quantify the kinase activity of Plk4 in the ternary complex (versus Plk4 alone) both *in vitro* and *in cells*.

Response: There is no well-established *in vitro* assay that can be used to measure Plk4 kinase activity. Plk4 *trans*-autophosphorylates both the β TrCP-binding S285 and T289 residues (Holland AJ, et al., 2012, *Genes & Dev*) and the activational phosphorylation sites (S698, S700, T704, and T707; the SSTT motif) (Park JE, 2019, *Nat Commun*). Therefore, to determine the effect of Vpr•VprBP on Plk4, we used a specific antibody against the p-SSTT motif (Park JE, 2020, *Cell Cycle*) and quantified the level of phosphorylated SSTT motif (p-SSTT) after performing *in vitro* kinase assays with GST-Plk4(1–836; Δ PB3), the longest Plk4 form available. The Plk4 binding-defective Vpr(1–79)•VprBP complex serves as a control. The results are provided in Fig. 4g and Supplementary Fig. 4p.

To determine the effect of Vpr on Plk4 kinase activity *in vivo*, we used HEK293T cells cotransfected with mCherry-Plk4, FLAG-STIL and mGFP-Vpr or mGFP-Vpr(1–79) for 10–12 h. Plk4-dependent STIL phosphorylation at the S1108 residue (Ohta M, et al., 2014, *Nat Commun*) was quantified using an anti-STIL p-S1108 antibody. The results are shown in Fig. 4h and Supplementary Fig. 4r. We found that Vpr, but not the Plk4 binding-defective Vpr(1–79), activates Plk4 approximately 2-fold in both *in vitro* and *in vivo* experiments. A modest elevation of Plk4 level (< 2-fold) is reported to be sufficient to induce centrosome amplification and aneuploidy and generate various spontaneous tumors in a mouse model (Levine, MS, et al., 2017, *Dev Cell*). Our new findings are described on page 7 (“Since Plk4’s ring-----”) and discussed on page 10 (“The approximately two-fold---”).

4. In Fig. 4d, the addition of MG132 is mentioned, but the reason for its inclusion is not provided. Please clarify the rationale behind the use of MG132. Additionally, it is essential to ascertain whether the cell cycles under each condition are comparable. Since ring-like localization of Plk4 is observed in early G1 phase, any alterations in the cell cycle duration resulting from the expression of Vpr could influence the analysis appropriately. Therefore, the authors should investigate whether the expression of mEGFP-Vpr affects the duration of G1 phase. Alternatively, the localization of Plk4 within the same cell cycle stage could be analyzed.

Response: Ohta et al (Ohta M, et al., 2014, *Nat Commun*) showed that treatment of cells with MG132 greatly enriches the population of cells with undegraded, ring-like Plk4 around a centriole. This is noted on page 12 by citing the Ohta et al. paper. In addition, MG132 treatment did not alter the cell cycle (Supplementary Fig. 4o).

To address the effect of Vpr-induced altered cell cycle on the level of Plk4’s ring vs. dot state, we performed two independent experiments. First, as the reviewer requested, we closely analyzed the cell cycle duration in G1 vs. S+G2/M (i.e., before vs. after centriole duplication) using cells stably expressing control vector, Vpr, or Vpr(1–79) and synchronously released from double thymidine block (detailed experimental scheme and raw FACS data are provided in the Source Data). Under our experimental conditions, Vpr induced an approximately 8% increase in the S+G2/M population (Supplementary Fig. 4m–o). Therefore, to minimize the effect of the altered cell cycle on Plk4’s ring vs. dot state, we also quantified the cells with multiple centrosomes after treating the cells with thymidine or nocodazole (Supplementary Fig. 4n,o). Under these conditions, the level of the S+G2/M population was similar among the three samples (see the table below).

% [S + G2/M]	mGFP	mGFP-Vpr	mGFP-Vpr(1–79)
Asyn	69.26	78.59	70.23
MG132	72.75	79.88	74.94
Thy	86.96	86.33	84.96
Noc	98.03	97.50	97.60

Taken together, these results solidify our view that Vpr can promote Plk4's ring-to-dot relocalization (Fig. 4d, graph) and induce multiple centrosomes independently of the cell cycle (Fig. 4f and Supplementary Fig. 4l,n,o). This interpretation is consistent with the data shown in Supplementary Fig. 4f that VprK27M, relieved of Vpr-imposed G2 arrest (Maudet C, et al., 2011, *J Biol Chem*), also induces a nearly undiminished level of cells with overduplicated centrosomes.

5. In the manuscript, the authors claim that Vpr and VprBP promote the dot-like localization of Plk4. However, the connection between the dot-like localization of Plk4 and centriolar overduplication is unclear. It has been hypothesized that the dot-like distribution of Plk4 restricts the potential sites of centriole overduplication and may even suppress centriolar overduplication. Therefore, comprehending their model is not straightforward. The authors should explain the potential mechanisms through which changes in Plk4 distribution lead to centriole overduplication.

Response: It is generally accepted that Plk4's catalytic activity-dependent ring-to-dot relocalization is critical to inducing Plk4-dependent centriole duplication (Yamamoto S and D Kitagawa, 2019, *Nat Commun*; Park JE, et al., 2019, *Nat Commun*; Yamamoto S and D Kitagawa; 2021, *Curr Opin Str Biol*). In addition, overexpression of Plk4 WT, not the kinase-inactive K41M, induces multiple centrioles (Kleylein-Sohn J, et al., 2007 *Dev Cell*; Park JE, et al., 2019, *Nat Commun*) by recruiting downstream components, such as STIL and Sas6 (Ohta M, et al., 2014, *Nat Commun*). Somewhat paradoxically, the Plk4 10A mutant lacking the 10 phosphorylatable residues within its linker 1 (L1) region can induce dramatic centriole overduplication, suggesting that L1 phosphorylation could suppress centriole amplification (Scott P, et al., 2023, *J Cell Biol*). However, it should be noted that the 10A mutant is at least several times more stable than the WT (Scott P, et al., 2023, *J Cell Biol*). Furthermore, the 10A mutant shows a substantially greater liquid-liquid phase separation (LLPS) activity than that of WT (Yamamoto S and D Kitagawa, 2019, *Nat Commun*). Condensation properties of Plk4 are thought to be critical for regulating centriolar copy number (Yamamoto S and D Kitagawa, 2019, *Nat Commun*). Therefore, the 10A mutant's increased protein amount and condensation activity could have induced centriole overduplication (note that the 10A mutant should be catalytically active and capable of phosphorylating STIL and recruiting Sas6). Here we showed that Vpr activates Plk4 approximately 2 fold (our new Fig. 4g and Supplementary Fig. 4p) without altering its stability (Supplementary Fig. 3j,k). Thus, we propose that Vpr-induced Plk4 activation potentiates the ability of Plk4 to undergo LLPS-mediated ring-to-dot relocalization and induce centrosome overduplication. This view is now stated in Fig. 6 legend.

6. Previous studies have reported the interaction between Vpr and Cep78 mediated by the binding of Vpr to VprBP (Hossain et al., 2018, *J Biol Chem*). It has also been shown that the expression of a Vpr mutant (Q65R) lacking interaction with VprBP and Cep78 fails to induce centriole overduplication (Hossain et al., 2018, *J Biol Chem*). Additionally, Cep78 has been implicated in Plk4-mediated centriole overduplication (Brunk et al., 2016, *J Cell Sci*). Thus, it appears that Cep78 plays a significant role in HIV-mediated centriole overduplication. To fully understand the mechanisms underlying HIV-induced centriole overduplication, the authors should consider the involvement of Cep78 and conduct experimental investigations to address this aspect.

Response: To address whether Cep78 is involved in the Vpr•VprBP•Plk4 complex-dependent centriole overduplication, we first examined the capacity of Cep78 to interact with VprBP compared to that of Plk4. Our results show that a substantial level of Cep78 coprecipitates with VprBP (an approximately 2-fold lower level when compared with that of Plk4) (new Supplementary Fig. 4d). Remarkably, however, depletion of Cep78 failed to influence the ability of Vpr to induce multiple centrosomes (new Supplementary Fig. 4e). These findings suggest that Cep78 forms a complex with VprBP and Vpr, as demonstrated previously (Hossain D, et al., 2017, *J Biol Chem*) and that this complex functions independently of the Vpr•VprBP•Plk4 complex. As this reviewer noted, Cep78 is shown to interact with the EDD•DYRK2•DDB1•VprBP complex, and this interaction appears to be important in maintaining CP110 stability and centrosome homeostasis

(Hossain, D, et al., 2017, EMBO Rep Vol p). Our data showed that Cep78 is not required for Vpr•VprBP•Plk4-induced centriole overduplication, although a minor role of Cep78 for this event cannot be eliminated. This new finding is described on page 6 (“On the other hand, ---“).

7. In Figure 4, the authors demonstrated that Vpr(1–79), which lacks interaction with VprBP and Plk4, does not induce centriole amplification. This finding suggests that the interaction between Vpr and Plk4 through VprBP promotes centriole overduplication. However, considering that the C-terminus of Vpr and VprBP may interact with other proteins, including Cep78 (Hossain et al., 2017, EMBO Rep), it is not appropriate to conclude that the Vpr-VprBP-Plk4 complex solely drives centriole overduplication. If feasible, it would be helpful to examine a Vpr variant with a point mutation that lacks interaction with Plk4 but can still interact with Cep78.

Response: This comment is related to the Comment #6. As mentioned above, we clearly demonstrated that depletion of Cep78 does not influence the ability of the Vpr•VprBP•Plk4 complex to induce centrosome overduplication. This finding strongly suggests that Cep78 is likely a component of a functionally independent VprBP-containing complex, such as EDD•DYRK2•DDB1•VprBP-Cep78-CP110 complex (Hossain, D, et al., 2017, EMBO Rep). Alternatively, Cep78 could be associated with the Vpr•VprBP•Plk4 complex, but its significance for the function of the complex is ignorable. In addition, whether Vpr directly interacts with Cep78 or interacts with Cep78 via VprBP is not known at present.

Minor points:

1. In Figure 4b, the authors did not include the quantification of centrosomal Plk4 intensity in cells expressing only mEGFP (without Vpr) as a control. It is important to determine whether the expression of mEGFP-Vpr affects the centrosomal intensity of Plk4 compared to the expression of mEGFP alone. Including this control would help assess the specific impact of mEGFP-Vpr on Plk4 localization.

Response: We included the mGFP + shLuc controls for both graphs in Fig. 4b.

2. In Fig.4b, a difference in the Plk4 intensity between control and shVprBP does not seem to be significant ($n=3$ or more than 3), compared to that in the Vpr intensity. It would be helpful if the authors could provide an explanation for this observation.

Response: Although the differences in the Plk4 intensities are not significant in cells with $n \geq 3$ centrosomes, the number of cells in this group is substantially higher in shLuc/mGFP-Vpr cells ($n=31/109$, 28%) than in shVprBP/mGFP-Vpr ($n=5/91$, 5%) and shPlk4/mGFP-Vpr ($n=3/90$, 3%) cells. To add clarity to these data, this information is now provided in the graphs.

3. Regarding the EM images shown in Figure 5a, centrioles 1 and 2 appear to be mature or elongated, while the other amplified centrioles seem to be shorter or immature. It would be beneficial if the authors could provide an interpretation of these observations. Do these findings reflect the nature of the centrosome amplification phenotype induced by GFP-Vpr expression? Additionally, it would be informative to know how the amplified centrioles in HIV-infected or patient cells compare to the ones observed in this study.

Response: It is reported that the expression of an activated Plk4 induces elongated centrioles (Park, JE et al., 2019, Nat Commun). In addition, there may be a local CP110 degradation, which leads to centriole elongation (Hossain D, et al., 2017, J Biol Chem). Addressing the reviewer’s question about the centriole morphology in HIV-1-infected or patient cells would be very challenging, considering the low percentage of cells with multiple centrosomes often scattered in the vast cytosolic space. A systematic analysis with a large number of TEM samples would be necessary to adequately address this question. Since our current study focuses on revealing the existence of the Vpr•VprBP•Plk4 complex but not the effect of Vpr on centriole structure, we think it would be best if this reviewer’s question could be addressed in another study.

4. In cells expressing GFP-Vpr, do the amplified centrioles have the ability to induce multipolar spindle formation during mitosis? Since centriole amplification can potentially lead to the formation of extra spindle poles, it would be relevant to investigate whether the amplified centrioles in GFP-Vpr-expressing cells contribute to multipolar spindle formation during cell division.

Response: As requested, the data showing the induction of multipolar spindles by mGFP-Vpr is now provided in the new Fig. 4f and new Supplementary Fig. 4l. Detailed information about how the quantification was performed is described in the legend.

5. The authors could examine whether the distribution of Plk4 around centrioles is affected in HIV-infected or patient cells. Investigating the impact of HIV infection on the localization and distribution of Plk4 around centrioles would provide valuable insights into the cellular mechanisms underlying centrosome amplification in the context of HIV infection or T-cell lymphoma in patients.

Response: As requested, we immunostained CEM-SS cells infected with HIV-1 and examined the localization patterns of Plk4 around a centriole. The results are provided in Supplementary Fig. 6c, along with the quantified data. As expected, if HIV-1 Vpr promotes Plk4-dependent centriole duplication, the level of the dot-state Plk4 is substantially increased in HIV-1-infected p24⁺ cells. Note that HIV-1-infected cells with more than three centrosomes exhibit a more pronounced dot-state Plk4. This finding is stated on page 8 (“by promoting Plk4 relocalization----”).

Reviewer #2 (Remarks to the Author):

In this manuscript by Park and Kim et al., the authors find that a small proportion of CD4⁺ T cells isolated from HIV-1 infected individuals display centrosome amplification. Interestingly, this phenomenon is absent in healthy control cells. Given the known association of Plk4 and centrosome duplication, the authors perform biochemical analyses to identify and characterise a ternary complex formed by Vpr-VprBP-Plk4. They also show that Vpr can induce multi-centromeres in model cell lines and primary CD4⁺ T cells. This is an interesting and important study and, overall, well performed. However, it could be strengthened to convince the reader of some of the phenotypes.

1. Figure 1 sets the scene for the paper, and some of the data from this figure are hard to interpret. The age of the HIV-1-infected individuals, their additional co-morbidities, and all details from the healthy control samples analysed were absent. It would be helpful to know how accurately the healthy controls matched the infected individuals. Additionally, the authors draw their conclusions after analysing two healthy controls.

Response: As requested, we provided all the clinical information available for 10 HIV-1⁺ individuals (a total of 14 samples with four paired samples obtained before and after ART). Also, we increased the number of healthy control CD4⁺ T cells to 14. Detailed information for healthy controls and HIV-1⁺ individuals is provided in Supplementary Fig. 1c–e. The healthy control cells are largely age-matched to the samples obtained from the HIV-1⁺ individuals (Supplementary Fig. 1f). The control CD4⁺ T cells showed less than 0.05% of overduplicated centrosomes (Fig. 1b and Supplementary Fig. 1c).

2. Given the low number of cells analysed (around 300 for HIV-1-positive individuals) and the modest percentage of cells with multi-centrosomes (around 1%) for 3/5 samples, the authors detected 3-6 cells with multi-centrosomes. Therefore, the power of these observations (combined with the lack of information about some of the samples) seems weak. These data are essential for the story's development, so it is important for the authors to strengthen these data by including additional healthy controls (clearly stating how they were matched to HIV-1-positive individuals). It would also be helpful to have additional samples from HIV-1-positive individuals at different stages of the disease. Control samples should be relatively easy to obtain, so a 3:1 ratio control:case would provide more confidence in the data.

Response: As requested, we analyzed additional healthy and patient samples with matching ages (see Comment #1 above)

We also analyzed four sets of samples purified from PBMCs obtained from four individuals before and after ART (note that it is very difficult to find before-ART samples). Importantly, although the number of paired samples is low, comparative analyses for these samples allowed us to determine the effect of ART on the HIV-1-induced centrosome amplification. These new results are provided in Fig. 1a,c and Supplementary Fig. 1d. We found that while ART significantly diminished the level of cells with overduplicated centrosomes, still an average of 26% of cells show overduplicated centrosomes. A study demonstrated that a substantial amount of Vpr (4 pg/mL of blood) remains long after ART (Agarwal N, et al., 2013, *Sci Transl Med*). Since Vpr is membrane-permeable (Sherman MP, et al., 2002, *Virology*; Gross DA, et al., 2019, *Sci Rep*), it seems plausible that Vpr's non-cell-autonomous function can contribute to the generation of cells with overduplicated centrosomes. This possibility is discussed on page 9 (“Studies show that Vpr---“). We agree with this reviewer that examining the non-cell-autonomous effect of Vpr (see Comment #3 below) will be potentially important to understanding the etiology of HIV-1-associated cancers other than T-cell lymphoma. However, exploring this possibility would require a new round of extensive studies with different cell types and new experimental approaches. Based on the Editor's opinion, we opted out of addressing this issue.

3. Importantly, the majority of lymphomas in patients with HIV are high-grade B cell lymphomas, plus (a minority of) Hodgkin's i.e. they do NOT arise from HIV-infected CD4+ T cells (some are related to herpesviruses, particularly EBV and HHV-8). In this paper most (possibly all) of the features they describe relate to cell-autonomous effects i.e. in principle, relating to HIV-infected CD4+ T cells. They address this issue here:

Response: See our Response to the Comment #2 above.

4. Therefore, showing that this might/could occur in uninfected cells would seem essential e.g. (as a minimum) Vpr-VLPs in T cells, or of more relevance B cells (this would have to be with "free" Vpr). Otherwise it's unclear how their interesting observation relates to the issue they wish to address – how HIV-infected individuals are more susceptible to a range of, predominantly B-cell derived lymphomas?

Response: This comment is related to Comment #3. We agree that demonstrating whether the effect of Vpr goes beyond HIV-infected CD4⁺ T cells would be very important and is likely the direction of our future research. Please note that in this study, we focused on demonstrating how T-cell lymphoma can be developed in patients living with HIV. As the Editor kindly advised us, for this manuscript, we focused on examining the effect of HIV-1 infection within CD4⁺ T cells. Further investigation with primary B cells with purified Vpr would require a series of new experiments to prove a non-cell-autonomous Vpr function. While exploring this direction will be worthwhile, we think it is beyond the scope of this manuscript. In the discussion, we noted that “Exploring this intriguing possibility could require a new layer of research.”

Somewhat related to this question (mentioned in this reviewer's comments below) is whether Vpr-VLP is sufficient to drive centriole overduplication. Using CEM-SS cells infected with HIV-1 and treated with 2 μM raltegravir for 38 h, we found that encapsidated Vpr appeared to be sufficient to induce centrosome overduplication. This new result is provided in Supplementary Fig. 6g,h and described on page 8 (“Remarkably, while treatment----“).

5. Are the abnormalities they describe actually seen in and/or characteristic of HIV-associated lymphomas (or solid cancers) – I couldn't find this information? They say they look at cells from 5 HIV positive donors, although details about these patients seemed to be absent e.g. what stage of infection, were they viraemic, on treatment, have advanced HIV, etc.

Response: It is well documented that centrosome amplification is common not only in hematological malignancies but also in solid cancers (see a review article, Chan JY, 2011, *Int J Biol Sci*). This is stated on page 10 (“Centrosome amplification, a cause of chromosomal instability-----”). Also, the requested patient

information is provided in Supplementary Fig. 1c–f. This covers all 14 samples obtained from people living with HIV-1 (four sets of them are matching samples obtained before and after ART).

Other points:

1. *The biochemical data presented in Figures 2 and 3 is convincing and well-performed.*

Response: Thank you.

2. *The microscopy data in Figures 4, 5 and, 6 is less convincing.*

Response: To improve the quality of the images, we generated all new figures with high-resolution three-dimensional structured illumination microscopy (3D-SIM) images. These new data (marked with red boxes) include Fig. 4d, Supplementary Fig. 4i–k (right, second panel), Supplementary Fig. 5b, and Supplementary Fig. 6i, 6j.

3. *In some figures, it is unclear which phenotype the authors are showing. For example, in Supplementary Figure 5A, the authors mention: “Vpr robustly induced centrosome overduplication and VprBP further augmented it (Fig. 5b and Supplementary Fig. 5a)”. However, I find it hard to see that in the images shown, and am concerned about the quantification.*

Response: As requested, we improved the quality of Supplementary Fig. 5b (Supplementary Fig. 5a in the 1st submission) by conducting a 3D-SIM imaging analysis. Cells with multiple centrosomes were quantified by carefully examining Cep152 signals in all the z-sections of each confocal microscopy image. This is a well-accepted laboratory practice.

4. *For all this section, it would help to include a CellTracker staining such as the one in Inanç et al., MBOC 2010. (In some way, in Supplementary Figure 6, p24 staining acts as a CellTracker and greatly facilitates the interpretation of the images).*

Response: As this reviewer suggested, we used p24 as a marker for HIV-1 infection for all the experiments carried out with TZM-bl, CEM-SS, and primary CD4⁺ T cells (Supplementary Fig. 6a–e, 6h, and 6j). This notion is clearly stated in all the pertinent graphs (noted as “among p24⁺ cells”) and figure legends.

5. *In figures showing mGFP-Vpr (e.g. 4D, 4E), adding lower magnifications, DIC, and DAPI is important. It is hard to understand which part of the cell is being shown. The difference in localisation between mGFP-empty and mGFP-Vpr (if any) is hard to see.*

Response: As requested, low-magnification images for Fig. 4d,e are provided in the new Supplementary Fig. 4i and 4j, respectively. In our 3D-SIM imaging system, which uses a single filter for all fluorescence signals, DAPI staining makes it difficult to locate signals from green and red channels. Therefore, DAPI staining was intentionally omitted. However, because of background mGFP signals and nuclear-localized mGFP-Vpr signals, we could easily discern nuclear morphologies, which were drawn with dotted lines in Supplementary Fig. 4i,j.

While control mGFP exhibited diffused signals, mGFP-Vpr showed localized signals around centrosomes (Supplementary Fig. 4i; we described it as a “nebulous” appearance in the text), which became more pronounced when cells were treated with MG132 (Supplementary Fig. 4k). The mGFP-Vpr(1-79) mutant exhibited somewhat less-distinguishable signals around centrosomes.

6. *The authors state: “Notably, mGFP-Vpr alone effectively localized to the pericentriolar material (PCM) region, often showing a “nebulous” appearance that encompasses multiple dot-like Plk4 signals (Fig. 4d and Supplementary Fig. 4f). In contrast, the Vpr(1–79) mutant localized poorly to centrosomes and failed to significantly induce the dot-like Plk4 signals (Fig. 4d and Supplementary Fig. 4f)”. However, this observation is not obvious when looking at the images displayed. An alternative interpretation of the images would be*

that they both poorly localise at the centrosomes or that Vpr(1–79) better localises at the centrosomes. Do authors have better representative images to show this phenotype?

Response: As we stated in our response to Comment #5 above, Vpr exhibited a somewhat broadly localized pattern surrounding centrioles. Treatment of cells with MG132 enhanced this signal (Supplementary Fig. 4i, 4j, and 4k). To better visualize these signals, the intensity of the green channel in Supplementary Fig. 4f (currently Supplementary Fig. 4k) is now adjusted. In addition, the second panel of the Supplementary Fig. 4k, right, is replaced with a better-representing image.

7. Why is the distribution of mCh-VprBP more cytosolic when co-expressed with mGFP-Vpr(1–79)? Is the change in the distribution of mGFP-Vpr when co-expressed with mCh-VprBP(1–1427) representative?

Response: Vpr localization does not change in the presence or absence of mCh-VprBP(1–1427). To better represent our data, we now provide 3D-SIM images to improve the quality of Supplementary Fig. 5a (current Supplementary Fig. 5b).

8. For Figure 5a, controls and quantification were not included. Additionally, the authors state: “The result showed that some cells exhibit up to 7 centrioles”. However, only one cell was analysed (according to the data shown). As is, these data would be better fit as supplemental information in Figure 4.

Response: We provided multiple immunostained data to show that mGFP-Vpr, but not control mGFP alone, induces multiple centrosomes (Figs. 4d,e, 6d, Supplementary Fig. 4i–k, and Supplementary Fig. 6i). The TEM image in Fig. 5 was provided to confirm that mGFP-Vpr indeed induces multiple centrioles. Another TEM image showing four centrioles is now provided in Supplementary Fig. 5a. We think providing a TEM image in the main text would be beneficial for the manuscript's clarity. Because centrioles are generally localized on different z-planes, serial sectioning, tomographic collection, and subsequent reconstruction of all images for each cell are required to reveal all the centrioles present in a cell.

9. Again, for Figure 6C, adding lower magnification, DIC, and DAPI is critical. Are the authors showing only a single cell?

Response: As requested, low magnification images for Fig. 6c (currently Fig. 6d) are now provided in the new Supplementary Fig. 6i. Multiple cells are shown in several figures, including Supplementary Fig. 6a–e,h,j.

10. Is Plk4 expressed in primary CD4+ T cells? Can the authors include this antibody in the western shown in Supplemental Figure 6D? (also, a small suggestion would be to rename “PBMCs CD4+ T cells” to “primary CD4+ T cells, to avoid confusion).

Response: Now the requested Plk4 immunoblot is provided (Supplementary Fig. 6i). The specificity of this antibody is shown in Supplementary Fig. 3l. As suggested, we renamed “PBMCs CD4+ T cells” to “primary CD4+ T cells”.

11. I couldn't find the information about the HIV-1 plasmids used in Figure 6. How did the authors reconstitute the coding sequence of Tat in the Vpr(1–79) mutant (overlapping orfs)? What percentage of the primary CD4+ T cells are infected for each condition? Are the cells shown in Figure 6C infected? Can the authors show they are infected? Is this data representative of multiple donors? For Figure 6D, were the cells first selected based on infection status?

Response: Additional information about constructing Vpr(1–79) is provided as requested. In brief, we generated a short fragment containing a stop codon immediately after the S79 residue and inserted it using the EcoRI and Sall sites [The Sall site is located before the S79 residue. Therefore, we add an extra ATAGCTGATGTCGAC sequence containing a stop codon (TGA) and a second SaII site (GTCGAC)]. The construct does not disrupt the Vif and Tat ORFs. The sequence information is provided.

ATGGAACAAGCCCCAGAAGACCAAGGGCCACAGAGGGAGCCATACAATGAATGGACACTAGAGCTTTTAGAGGAACT
TAAGAGTGAAGCTGTTAGACATTTTCCTAGGATATGGCTCCATAACTTAGGACAACATATCTATGAAACTTACGGGGAT
ACTTGGGCAGGAGTGGAAGCCATAATAAGAATTCCTGCAACAACCTGCTGTTTATCCATTTTCAGAATTGGGTGTCGACATA
GCTGATGTCGACATAGCAGAATAGGCGTTACTCGACAGAGGAGCAAGAAATGGAGCCAGTAGATCCTAG [the EcoRI
site (GAATTC) and two Sall site (GTCGAC) are underlined; a STOP codon TGA is marked].

12. Many of these questions are aimed at understanding if the authors believe this is a cell-autonomous or non-cell-autonomous effect. Vpr is packed in virions at equimolar ratios with Gag. Can Virus-like Particles (VLPs) packed with Vpr reproduce the effects seen?

Response: This comment is related to Reviewer #2's Comment #4 and Reviewer#4's Comment #2. As addressed above, CEM-SS cells infected with HIV-1 and treated with 2 μ M of raltegravir for 38 h exhibited an approximately 2-fold diminished level of overduplicated centrosomes (the results are shown in the new Supplementary Fig. 6g,h). Since a low level of p24 was sustained even after treating the cells with 20 μ M of raltegravir, we analyzed the effect of raltegravir at 2 μ M (Supplementary Fig. 6g,h). Our data suggest that encapsidated Vpr is sufficient to induce a significant degree of centrosome overduplication. This new data is provided in Supplementary Fig. 6g,h and described on page 8 ("Remarkably, while treatment of CEM-SS---"). Based on the editor's assertion that examining the effect of Vpr in a non-self-autonomous system is not required, we focused on determining Vpr function in CD4⁺ T cells.

13. Could they make a longitudinal analysis for some of the samples analysed?

Response: This comment is related to Reviewer #2's Comment #2. We were able to obtain four pairs of samples obtained before and after ART. Our detailed response is provided above. Briefly, the results show a significantly (74%) diminished level of centrosome overduplication after ART (Fig. 1a,c and Supplementary Fig. 1d).

14. In the discussion, the authors say: "In addition, since a substantial level of Vpr (10 pg/mL–10 ng/mL of blood) is present in the blood of people living with HIV-1", 68-70. It could be important to mention that Vpr levels correlate with RNA levels 69 and that ARV should bring both factors to undetectable levels.

Response: A previous study showed that while ART-treated HIV-1+ people showed undetectable viral load, they still exhibited a substantial level (3.9 pg/mL) of Vpr in their blood. Based on this observation, they concluded that Vpr produced by HIV-1 persisting in reservoirs can circulate in the blood of ART-treated HIV patients with undetectable viral load (Agarwal N, et al., 2013, *Sci Transl Med*). We discuss this point on page 9 ("Studies show that----").

Reviewer #3 (Remarks to the Author):

In this study, Park and colleagues investigate the mechanisms underlying previous findings indicating that HIV-1 infection and notably the virus-encoded Vpr accessory protein, disrupts centrosome homeostasis, a condition that may contribute to oncogenesis by driving aneuploidy. They provide evidence that CD4+ T lymphocyte isolated from the blood of persons living with HIV (PLWH) display to varying extent overduplicated centrioles although as discussed below it is difficult to assess the specificity of this effect as these observations were made on the total CD4+ T cell population (HIV-positive and HIV negative) and it is unclear if the two healthy controls were age-matched with the five PLWH. Consistent with previous studies, they show that HIV-1 Vpr induces centrosome amplification; importantly, they uncover a novel mechanism used by Vpr to hijack the centriole duplication machinery and induce centrosome amplification and aneuploidy. Using biochemical and cellular biology analyses, they provide novel and strong evidence that Vpr forms a cooperative ternary complex with VprBP, the substrate recognition receptor of the CUL4-DDB1 E3 ubiquitin ligase complex, and the Polo-kinase-like 4 (Plk4), a key regulator of centriole biogenesis that is found to form a complex with VprBP in normal conditions. Indeed, the authors provide evidence that the formation of a ternary complex with Vpr does not lead to Plk4 ubiquitination and degradation but rather enhance Plk4

functionality by driving its relocalization to the procentriole assembly site. By defining and characterizing the domains of Vpr, VprBP and Plk4 that are involved in the formation of this ternary complex they further demonstrate that the loss of complex formation abrogates Vpr ability to induce centriole over-duplication. To underline the biological significance of their findings, they show that WT HIV but not Vpr mutants that are unable to form a ternary complex with VprBP and Plk4, induce centrosome amplification and aneuploidy in a fraction of infected cells in various cellular model of HIV infection, including primary CD4+ T cells.

Overall, this is an interesting, and well executed study that uncover a novel host target of Vpr (Plk4) and a novel mechanism by which this viral encoded accessory protein hijacks the centriole duplication machinery, inducing centriole over-duplication and aneuploidy, independently of its ability to induce a cellular growth arrest at the G2 stage of the cell cycle. While the biochemical characterization of the Vpr-VprBP-Plk4 complex and the impact of various mutants on Vpr capacity to disrupt centrosome biogenesis represent strong components of the study, the extrapolation of these observations obtained in model systems in vivo remains rather weak. Additionally, while aneuploidy can be a driver of oncogenesis, the authors do not address the apparent paradox of Vpr mediating such a process but inducing at the same time a cellular growth arrest and apoptosis in HIV-infected cells.

Specific points:

1-Figure 1 and supplementary figure 1. Authors detect more duplicated centrioles (1-5%) within the total CD4+T cells from people living with HIV when compared with normal healthy donor. It is unclear from the information provided if the healthy donor were age-matched as age could influence the comparison. It seems also that the frequency of centriole duplication is not linked to viral load. as observed in patient 3. Was the number of years living with HIV considered in the analysis? In the absence of evidence that the centriole over-duplication is occurring primarily in HIV+ CD4+ T cells it is difficult to assess whether these observations are specific to HIV infection or other confounding effects.

Response: The age profiles of the 12 healthy individuals (2 are unknown) and ten individuals with HIV-1 are provided in Supplementary Fig. 1f. They are largely age-matched with similar median ages (Supplementary Fig. 1f). All the clinical data for 14 healthy individuals and ten patients with HIV-1 are provided in Supplementary Fig. 1c-e.

As the reviewer alluded, it is important to determine whether the frequency of centriole duplication is linked to viral load. With the five HIV-1⁺ samples, it is difficult to draw any meaningful conclusion because of the many variables among different individuals. Therefore, we analyzed four pairs of samples obtained before and after ART from the same HIV-1⁺ individuals. Results provided in the new Fig. 1a,c and Supplementary Fig. 1d show that ART diminished the level of centrosome overduplication to approximately 26% of that observed with the cells obtained before ART.

Following the acute phase of HIV-1 infection, viral load decreases substantially. Because of the low number of cells infected with HIV-1 (approximately one out of 1,000 cells), it is difficult to locate infected primary CD4⁺ T cells from HIV-1⁺ patients. In addition, available antibodies against p24 and Vpr do not provide reliable sensitivity to locate HIV-1-infected primary CD4⁺ T cells directly by immunostaining. However, our additional analyses with four pairs of CD4⁺ T cells purified before and after ART from the same patients show that ART greatly diminishes the level of centrosome overduplication (see our detailed Response to Reviewer #2 Comment #2 above) and that HIV-1 causes centrosome overduplication in CD4⁺ T cells. The new data in Fig. 1a,c, and Supplementary Fig. 1d further support our original conclusion. In addition, our data in Fig. 6 and Supplementary Fig. 6 showed that in three CD4⁺ cultured cells (TZM-bl, CEM-SS, and primary CD4⁺ T cells), HIV-1 WT, but not the Plk4 binding-defective HIV-1 Vpr(1-79) mutant, specifically induces centrosome overduplication.

In addition, the reviewer brought up an interesting point (in the general review section above) regarding how Vpr, which induces a cellular growth arrest and apoptosis, can induce aneuploidy, a driver of

oncogenesis. We agree with the reviewer that Vpr-induced G2 arrest alone may do little to promote oncogenesis. However, Vpr is shown to cause accumulation of damaged DNA and subsequent genomic instability (Laguetta N, et al., 2014, Cell). The accumulation of mutations in oncogenes and tumor suppressors is necessary to bypass the Vpr-induced G2 arrest and resume uninhibited cell proliferation (Magrath I, 1992, Cancer Res). As suggested, this view is discussed on page 9 (“In addition, the accumulation of----”).

2-Figure 2: panel 2b Except for the NTD Flag- PLK4, and a very weak signal for CPB Flag-PIK4, no other Plk4 mutants are detected in the input fraction. Is there a difference in transfection efficiency between the different constructs? Moreover, the IP for the full length Plk4, reveals multiple bands below 116kDa. Are there multiple Plk4 isoforms that could interact with VprBP?

Response: The difference in the expression levels in the ligands may likely be due to the different expression levels or stability for each construct. The anti-FLAG bands smaller than the full-length Plk4 may likely be degradation products. Both phenomena are common in co-IP experiments.

3-Supplementary Figure 3b: this figure is aimed at demonstrating that Vpr-enhancement of Plk4 association to VprBP is not dependent on Plk4 catalytic site. Why perform a Plk4 pull down and not a VprBP pull down as shown in figure 3b to clearly show the enhanced association of Plk4 to the complex in the presence of Vpr. As it is the enhanced recruitment of Plk4 WT and mutant is not obvious.

Response: It is reported that when compared with Plk4 WT, catalytically inactive Plk4 shows > 10-fold higher expression level (Holland AJ et al., 2010, JCB). This is because Plk4, which functions as a homodimer, generates its *trans*-autophosphorylation-dependent suicidal degron motif recognizable by β TrCP (see additional information in our Response to Reviewer #1’s Comment #1). We found that it is much harder to compare co-IP efficiency if the input levels are so different. The current coIP scheme allowed us to compare the signal intensities between WT and its kinase-inactive form.

4-Figure 6A: although there is a significant increase in the number of centrosomes in HIV-1 Vpr WT infected cells, there is also an increase in the number of cells harboring 3 to 4 centrosomes in Vpr (1-79) and Vpr (-) infected cells that seems significant when compared to uninfected cells. Could the authors explain why in absence of Vpr but during HIV infection, there seems to be a deregulation in centrosome number?

Response: We consistently found that when compared with control uninfected cells, a small but significant fraction of TZM-bl and CEM-SS cells infected with HIV-1 Vpr(1–79) or HIV-1 Vpr(-) exhibit multiple centrosomes (Fig. 6a and Supplementary Fig. 6a,b; *P* value is now provided for clarity). These observations suggest that factor(s) other than Vpr could also induce centrosome overduplication at a low degree. This possibility is stated in the Supplementary Fig. 6b legend (“Note that, like the TZM-bl cells----”).

5-Figure 6: the authors show the p24 levels by western blot in the supplementary figure 6d and state that approximately 10% of the CD4+ T cells were infected. Figure 6C reveals that approximately 20% of p24+ cells display centrosome amplification, suggesting that about 2% of infected primary CD4+ T cells exhibit multi-centrosomes. However, analysis of chromosome numbers in HIV WT-infected primary CD4+ T cells indicate a reduction of 20% of total CD4+ T cells carrying a normal number of chromosomes and a concomitant increase in the frequency of cells carrying 43 , 44 , 45 chromosomes (less than 10% for each) . Based on these numbers, it seems that evidence of aneuploidy is detected beyond infected cells as well as cells showing centrosome amplification. Could the authors comment?

Response: The reviewer is correct in that the fraction of cells with aneuploidy appears much greater than that of cells with discernable p24 signals. This could be due to multiple reasons. One possibility is that, as this reviewer and reviewer #2 alluded, Vpr effect may go beyond the cell-autonomous system. It is already known that a substantial amount of Vpr (10 pg/mL–10 ng/mL of blood), which is cell membrane-permeable (Sherman MP, et al., 2002, Virology; Gross DA, et al., 2019, Sci Rep), is present in the circulating blood of HIV-1

patients (Levy DN, et al., 1994, *Proc Natl Acad Sci*; Hoshino S, et al., 2007, *AIDS Res Hum Retroviruses*; Agarwal N, et al., 2013, *Sci Transl Med*). We agree that investigating Vpr's role beyond a cell-autonomous system would be an important direction for future research. This is mentioned on page 9 ("Studies show that Vpr is---"). Based on the Editor's notion above, here we focused on investigating the significance of the Vpr•VprBP•Plk4 complex within its system.

Another possibility is that, since CD4⁺ T cells undergo 2- to 6-fold faster cell proliferation under HIV-1 infected conditions (Lempicki RA, et al., 2000, *Proc Natl Acad Sci*; Kovacs JA, et al., 2001, *J Exp Med*), these cells may have a compromised cell cycle checkpoint, as suggested previously (Baek, KH, et al., 2003, *J Leukoc Biol*). This possibility is stated on page 8 ("This could be in part-----").

Minor Points

1- Figure 3d: the Vpx variant used should be specified in the figure legend

Response: This information is now provided.

2- Discussion : The authors refer to several studies suggesting that PLWH are at greatly increased risk of developing in T cell lymphomas. Does the prevalence of these HIV-associated cancers decreases upon antiretroviral therapy? This would be important to discuss in light of the author hypothesis that HIV-encoded factors might be involved.

Response: This is a great suggestion. As the reviewer suggested, we cited a paper and added a phrase stating that "the incidence of NHL and other AIDS-defining cancers becomes drastically reduced during the ART period,-----" on page 9.

Reviewer #4 (Remarks to the Author):

The authors found that the HIV-1 accessory protein Vpr hijacks the centriole duplication mechanism, causing centrosome amplification and aneuploidy. The author claimed that Vpr could form a cooperative ternary complex with an E3 ligase component, VprBP, and polo-like kinase 4 (Plk4) and that the complex improved Plk4 functioning by encouraging relocalization to the procentriole assembly and inducing centrosome amplification. This is an intriguing concept; however, other aspects of the document contradict previous Vpr literature and are difficult to explain. Overall, there are significant flaws that, in my view, render this research inappropriate for *Nat Comm* at this stage. Specific remarks are provided below.

Major

1. The Fig. 1 is unconvincing. To infer that HIV-1 generates over-duplicated centrosomes in CD4⁺T cells in untreated HIV-1-infected persons, they must assess the levels of centrosome over-duplication in CD4⁺T cells from ART-treated patients since their viruses are inactive. According to the authors' premise, they should not find over-duplicated centrosomes in CD4⁺T cells from ART patients.

Response: As suggested, we examined centrosome abnormalities using four pairs of CD4⁺ T cells obtained before and after ART from as many individuals. The data are provided in Fig. 1 (The old figure is now placed in Supplementary Fig. 1, along with other additional samples we analyzed). The results show that the fraction of CD4⁺ T cells exhibiting overduplicated centrosomes is significantly diminished. Yet, the ART did not eliminate the number of cells with abnormal centrosomes even though the level of HIV-1 RNA remains very low (Fig. 1c and Supplementary Fig. 1d; compare before and after ART). It has been reported that while ART drastically reduces the risk of NHL and other AIDS-defining cancers, the risk of NHL in individuals with HIV-1 remains 9-fold higher than that in the general population (Hleyhel M, et al., 2013, *Clin Infect Dis*). In addition, a substantial amount of Vpr (4 pg/mL of blood) remains long after ART (Agarwal N, et al., 2013, *Sci*

Transl Med) (see our detailed Response to Reviewer #2 Comment #2). This is stated in the discussion on page 9 (“Studies show that Vpr is present---”) and in Supplementary Fig. 1d, legend.

2. The figures in this paper (Figs. 2, 3, 4, and 5) are not physiologically relevant to HIV-1. At the very least, the authors should repeat certain crucial T-cell tests. Show how Vpr-VprBP-Plk4 interact in CD4+T cells. It is a crucial experiment for us to assume that this has physiological significance. Finally, the authors proved in Fig. 6 that HIV-1 infection increases centrosome duplication. Unfortunately, this does not persuade. They should utilize HIV-1 infection, whether EFV is present or not. Please do not use HIV-1 that has not been infected as a control. Please confine HIV-1 infection to a single cycle as well.

Response: To demonstrate the Vpr-VprBP-Plk4 interaction *in vivo*, we performed two independent coimmunoprecipitation experiments—one with CEM-SS cells infected with HIV-1 and the other with HEK293T cells infected with lentiviruses expressing FLAG-Vpr. After immunoprecipitating Plk4 with an anti-Plk4 antibody, we detected coimmunoprecipitated Vpr in both cases. The results are provided in the new Fig. 6c and Supplementary Fig. 6f.

Also, we examined the effect of raltegravir (obtained from the NIH HIV Research Program), an integrase inhibitor (*Anker M and Corales RB, Expert Opin Investig Drugs, 2008, 17:97*). CEM-SS cells treated with various concentrations of raltegravir (2–20 μ M) showed that 2 μ M was sufficient to block most of the integrase activity (judging from the diminished level of p24) (Supplementary Fig. 6g). In the subsequent experiment, we found that CEM-SS cells treated with 2 μ M of raltegravir for 38 h exhibited an approximately 2-fold diminished level of overduplicated centrosomes (the results are provided in the new Supplementary Fig. 6h). Efavirenz (obtained from the NIH HIV Research Program), a reverse transcriptase inhibitor, was not used because, unlike raltegravir, treatment of CEM-SS cells with 0.3 μ M of efavirenz for 38 h induced overduplicated centrosomes in approximately 1.6% of the cells even without HIV-1 infection.

For all HIV-1-infected experiments, we used an *env*-deleted variant of pNL4-3, pNLenv1. Detailed information is provided on page 16 (“The *env*-deleted variant of pNL4-3-----”). Due to the absence of *env*, the infection is limited to a single cycle.

3. Why did the author investigate HIV-2 Vpx? This is quite unusual! Vpr is also found in HIV-2 and SIV! For this assumption, please erase the Vpx data and compare HIV-2 and SIV Vpr to HIV-1 Vpr. Furthermore, they should employ Vpr proteins from multiple HIV-1 strains (NL4-3, AD8, 89.6) to back up their findings.

Response: As this reviewer noted, we comparatively analyzed the ability of Vpx to form a ternary complex with VprBP and Plk4. This is because the structure of Vpx (*Wu Y, et al., 2016, Nat Str Mol Biol 23:933*) is reported to be similar to that of Vpr (*Morellet N, et al., J Mol Biol 2003, 327:215*) (this was noted in our original manuscript, page 3). We think keeping the Vpx data would be beneficial.

As suggested, we included HIV-2 Vpr and also included another HIV-1 Vpr variant (89.6) along with the two HIV-1 Vpr variants (NL4-3, the most prevalent form, and Lai) already described in our original manuscript (Supplementary Fig. 3e,f). We opted out SIV Vpr, because this study focuses on human HIV-1. We carried out additional coimmunoprecipitation analyses to demonstrate that both HIV-2 Vpr and HIV-1 Vpx failed to form a cooperative ternary complex with VprBP and Plk4, although a low level of Plk4 was present in the coprecipitates (new Supplementary Fig. 3g). This new finding is described on page 6 (“Unlike HIV-1 Vpr (NL4-3 and 89.6), HIV-2 Vpr----”). Consistent with these results, we also found that while HIV-1 Vpr(NL4-3) and Vpr(89.6) induce multiple centrosomes effectively (proportional to their expression levels; see Supplementary Fig. 4g, left), HIV-2 Vpr induced them only weakly (Supplementary Fig. 4g, right).

4. Please investigate whether or not the virion-associated Vpr generates this behavior. There is no generated Vpr with raltegravir during HIV-1 infection, just encapsidated Vpr. Vpr is encapsulated in HIV particles, and the author must determine which Vpr induces centrosome amplification.

Our response to this reviewer’s Comment #2 above addresses this question.

5. *Vpr does not seem to have a profile that promotes HIV-1 infection in CD4+T cells (PMID: 29669271), but it does increase HIV-1 infection in primary macrophages. The authors should additionally investigate if Vpr binds to PLK4 in non-proliferating macrophages (no centrosome?). This might be one reason for Vpr's different involvement in CD4+T and macrophages.*

Response: As the reviewer mentioned, Vpr has been shown to increase HIV-1 infection in primary macrophages. In addition, multiple studies showed that Vpr-induced G2 arrest provides a selective advantage for the virus to increase both its replication and gene expression in primary T cells and *in vivo* (Goh WC, et al., 1998, *Nat Med*; Gummuluru S and Emerman M, 1999, *J Virol*).

This reviewer suggested testing the Vpr-Plk4 interaction in macrophages. However, the primary focus of this study is investigating the consequence of forming the ternary complex in CD4⁺ T cells and its potential role in contributing to the development of T-cell lymphomas. Based on the editor's assertion that experiments in a non-self-autonomous system are not required, we opted out of addressing this comment. Nevertheless, we do agree with this reviewer that this is a direction that we can pursue in the near future.

6. *Fig. 4. Plk4 kinase has specific inhibitors! Please use these inhibitors to see whether kinase activity is required for HIV-1-induced over-duplication of centrosomes. This is critical.*

Response: As suggested, we examined the consequence of pharmacologically inhibiting the Plk4 kinase activity using HIV-1-infected CEM-SS cells. Our data showed that treatment with 200 nM of centrinone (Wong YL, et al., 2015, *Science*) greatly diminishes the fraction of cells with multiple centrosomes. The results are provided in the new Supplementary Fig. 6e. This result confirms the data in Fig. 4c and Supplementary Fig. 4c that show that depletion of Plk4 by RNAi greatly diminishes the level of Vpr-induced centrosome overduplication in U2OS cells.

7. *Please provide MOI for HIV-1 infection; it is critical to determine whether the phenotype is physiological or not.*

Response: Experiments with TZM-bl and CEM-SS cells were performed at the MOI of 1 [The MOI was determined using an HIV-1 p24 ELISA kit (Abcam) and TZM-bl cells]. Experiments with primary CD4⁺ T cells, which are difficult to infect, were carried out after infecting the cells with tenfold more HIV-1. This is stated on page 18 ("The multiplicity of infection-----").

Minor:

1. *From Fig. 1, we can observe that there were no multi-centrosomes in the healthy case group. However, it is unclear if the cells in this image were infected with HIV-1, despite the fact that not all cells in HIV patients were infected with the virus. More evidence is required.*

Response: The reviewer is correct in that only a small fraction (approximately 0.1%) of CD4⁺ T cells are infected with HIV-1. A low frequency of HIV-1-infected cells makes it challenging to locate them. In addition, currently available antibodies against p24 or Vpr do not yield reliable signals to locate HIV-1-infected CD4⁺ T cells directly from patient samples. Therefore, to improve the validity of our data in the original manuscript, we now provide data obtained from 14 healthy individual samples and 14 HIV-1 patient samples (among them, four sets of them are paired samples obtained before and after ART) with matching ages. We found that, in all cases, primary CD4⁺ T cells from healthy individuals showed less than 0.05% of them with multiple centrosomes (new Fig. 1a,b and Supplementary Fig. 1a,c). In contrast, primary CD4⁺ T cells from HIV-1 patients exhibited an average of 0.9–5.1% of multiple centrosome-containing cells (new Fig. 1a,b and Supplementary Fig. 1a,d,e). More importantly, comparative analysis performed with the four paired samples obtained before and after ART showed that ART effectively diminishes the level of overduplicated centrosomes (new Fig. 1a,c and Supplementary Fig. 1d). This result is described on page 4 ("Remarkably, all four samples----").

2. Based on Fig. 1, the author hypothesized that Plk4 was unregulated during HIV-1 infection. Did they examine other plausible mechanisms, such as “accidental” activation of CDK2/cyclin E, which causes centrosome re-duplication and, perhaps, centrosome amplification? Is it feasible that additional PLK family proteins are involved? In essence, the centrioles are “primed” for duplication in the early G1 phase by pericentrin cleavage, which leads to centriole disengagement. The cyclin-dependent kinase 1 (CDK1), polo-like kinase 1 (PLK-1) and Aurora kinase A control this process.

Response: We cannot eliminate other kinases contributing to the HIV-1-induced centrosome overduplication. However, our data show that Plk4 depletion almost eliminated Vpr-induced centrosome overduplication (Fig. 4c and Supplementary Fig. 4c) and that TZM-bl, CEM-SS, and primary CD4⁺ T cells infected with HIV-1 WT, but not the Plk4 binding-defective HIV-1 Vpr(1–79) mutant, failed to induce centrosome overduplication. Moreover, treatment of HIV-1-infected CEM-SS cells with 200 nM centrinone, a Plk4-specific inhibitor (Wong YL, et al., 2015, *Science*), almost annihilated centrosome overduplication (new Supplementary Fig. 6e). These data strongly suggest that the Vpr•VprBP•Plk4 complex that we described here is largely responsible for the centrosome amplification induced by HIV-1. An earlier report shows that centrosome overduplication does not occur in cells arrested in G2 by irradiation (Watanabe N, et al., 2000, *Exp Cell Res*), suggesting that G2 arrest itself does not cause this phenotype. This is stated on page 6 (“This suggests that Vpr can induce-----”).

3. The author performed extensive IP experiments to illustrate the connection between PLK4 and VprBP in Fig. 2. Cell lines and exogenous transfection were used in all of these studies. There is no evidence for an interaction between endogenous PLK4 and VprBP, particularly in HIV-1-vulnerable target cells.

Response: See our response to Comment #4 below.

4. Similar questions were also included in Fig. 3. What is the endogenous level of the PLK4-VprBP-Vpr complex, particularly in HIV-1-vulnerable target cells? Is their interaction at HIV-1 Vpr+/- compatible with that seen in the cell line?

Response: As suggested by this reviewer, we examined whether Plk4 forms a complex with VprBP and Vpr *in vivo* by carrying out two independent coimmunoprecipitation analyses—one with CEM-SS cells infected with HIV-1 and the other with HEK293T cells infected with lentiviruses expressing Vpr. The results are provided in Fig. 6c and Supplementary Fig. 6f. These data further strengthen our discovery that HIV-1 Vpr hijacks the cellular VprBP•Plk4 complex and deregulates Plk4-dependent centrosome duplication machinery.

5. The author employed the HIV-2 Vpx protein as a negative control for VprBP binding in Fig. 3d, but they neglected one essential point. Vpr is widely known to be present in both HIV-2 and SIV. However, the author did not investigate whether HIV-2 and SIV Vpr could bind to VprBP. They did not even say if Vpr from various species and genera may trigger centrosome amplification.

Response: This is the same comment as the Comment #3 above. Please see our response to this reviewer’s Comment #3 above.

6. The author only investigated the impact of PLK4 in Fig. 4, but this does not completely describe whether Vpr of other proteins in the PLK family have the same effect on it.

Response: Among the four members of the Polo-like kinase subfamily, Plk4 plays a key role in inducing centriole duplication (Habedanck R, et al., 2005, *Nat Cell Biol*). Other data show that Plk1 is required for centriole elongation and centriole disengagement (Shukla A, et al., 2015, *Nat Commun*). Because HIV-1 infection induces centrosome overduplication in CD4⁺ T cells from people living with HIV-1, we focused on exploring a potential connection between Plk4 and overduplicated centrosomes. It is possible that Plk1 function, such as centriole disengagement, can also be influenced by HIV-1 either directly or indirectly. This could be another direction for future study.

7. The Figs. 4d and 4e are likewise vulnerable. Will HIV-2/SIV Vpr trigger centrosome amplification as well? Even the auxiliary protein Vpx, which does not bind to VprBP?

Response: This comment is also related to the Comment #3 above. We found that various HIV-1 Vpr variants (NL4-3 and 89.6) interact with VprBP and Plk4 well (Supplementary Fig. 3e,f and new Supplementary Fig. 3g). HIV-2 Vpr coprecipitates with VprBP and Plk4, but only weakly (Supplementary Fig. 3g). Ability of these Vpr variants to induce multiple centrosomes was largely proportional to the level of their capacity to generate the ternary complex (Supplementary Fig. 4g). HIV-2 Vpr induced multiple centrosomes only at a low level, even though it was expressed at a few-fold higher level than HIV-1 Vpr (NL4-3 or 89.6). This is stated on page 6 (“Unlike HIV-1 Vpr-----“). Vpx, which does not bind to Plk4, was excluded from this analysis. Also, we focused on HIV-1-encoded proteins to understand the relationship between HIV-1 and human cancers.

8. Although the author detected centrosome amplification in activated CD4+T cells and U2OS cell lines, further controls are needed to clarify the critical function of Vpr, particularly in HIV-2/SIV.

Response: As suggested, we included HIV-2 Vpr as an additional control. The results in Supplementary Fig. 3g and 4g show that HIV-2 Vpr, expressed 3.5-fold more than HIV-1 Vpr (NL4-3 and 89.6), induced only a mild impact on Plk4-dependent centriole duplication. We did not include SIV Vpr for our analysis because our studies focus on HIV-1-associated human disorders.

9. It is remarkable that PLK4 is expressed poorly in Supplementary Figs. 6b and 6c, despite the fact that it plays a critical role in centrosome amplification, as the author of the research pointed out. Is this due to the action of Vpr on other host proteins, which causes centrosome amplification?

Response: This is related to the Minor Comment #2 above. It is well documented that Plk4 is a master kinase regulating centrosome duplication. Because cells need precisely two centrosomes to divide, a low level of Plk4 expression is thought to be important to prevent centriole overduplication (unless perturbed by HIV-1 infection, etc.) and maintain genomic stability (*Nigg EA and AJ Holland, 2018, Nat Rev Mol Cell Biol*). Our data show that depletion of Plk4 by RNAi or pharmacological inhibition of Plk4 almost eliminated Vpr-induced centrosome overduplication (Fig. 4c and Supplementary Figs. 4c and 6e). In addition, we demonstrated that the Vpr•VprBP•Plk4 complex, but not the Plk4-binding defective Vpr(1–79)•VprBP•Plk4 complex, induces Plk4-dependent centrosome overduplication.

As reviewer #1 suggested, we examined whether a centrosomal protein, Cep78, which is reported to interact with VprBP and Plk4 (*Hossain, D, et al., 2017, EMBO Rep; Brunk, K et al., 2016, JCS*), is required for Vpr•VprBP•Plk4 complex-mediated centrosome overduplication. We found that although Cep78 somewhat weakly interacts with VprBP (Supplementary Fig. 4d), depletion of Cep78 did not influence the ability of Vpr to induce multiple centrosomes (Supplementary Fig. 4e). Therefore, we conclude that Cep78 is not required for the Vpr•VprBP•Plk4 to induce multiple centrosomes. Cep78 likely forms a complex with VprBP, but this complex is functionally distinct from the Vpr•VprBP•Plk4 complex.

10. The authors stated that VprBP and PLK4 were both required for Vpr to generate numerous centrosomes. However, VprBP expression was substantially higher than PLK4, and the shift in PLK4 expression following PLK4 was not visible in Supplementary Fig. 4. VprBP experienced the same thing. Endogenous expression of VprBP and PLK4 must be reevaluated.

Response: This question is related to the Minor Comment #9 above. Quantification of Plk4 signals in Supplementary Fig. 4c showed that approximately 90% of Plk4 is depleted by shPlk4 (quantified numbers are provided for easy comparison). VprBP depletion was nearly complete. All immunoblottings were performed with endogenous proteins.

The reviewer thinks that the level of Plk4 expression should change proportionally to that of VprBP. However, VprBP is not the only protein regulating Plk4 stability *in vivo*. Like our response to Comment #9 above, Plk4 copy numbers are tightly controlled in cells [only approximately 50 copies per centrosome

(Bauer M, et al., 2016 EMBO J) and approximately 4,000 copies per cell (Holland AJ, et al, 2012, Genes & Dev; Note that 4,000 will yield a very low concentration, given that the cytosol is approximately 100,000 times larger in volume than the centrosome space)]. The way that cells regulate Plk4-dependent centriole biogenesis is through *trans*-autophosphorylation-dependent liquid-liquid phase separation (Park JE, et al., 2019, Nat Commun; Yamamoto S and D Kitagawa; 2019, Nat Commun) and ring-to-dot relocalization around a centriole through a process called, symmetry breaking (Takao D, et al, 2019, J Cell Biol; Yamamoto S and D Kitagawa; 2021, Curr Opin Str Biol). Therefore, the way that Plk4 is regulated is very different from various other proteins.

11. *Supplementary Fig. 6d: The immunoblots must contain a PLK4 (anti-PLK4) blot to ensure that all proteins are expressed at similar levels.*

Response: Now the requested anti-Plk4 blot is provided.

12. *Please also investigate if Plk1 binds to Vpr since Plk1 is a member of the PLKs. Perhaps a nice Plk4 control.*

Response: As suggested, we performed a coimmunoprecipitation analysis with Plk1. We found that Plk1 weakly (approximately 3-fold weaker than Plk4) interacted with VprBP. Importantly, however, Vpr failed to augment this interaction (Supplementary Fig. 3h). Whether the VprBP-Plk1 interaction works in conjunction with the VprBP-Plk4 interaction to promote the Vpr•VprBP•Plk4-mediated centriole duplication remains to be further investigated. However, please note that depletion of Plk4 by RNAi or pharmacological inhibition of Plk4 almost eliminated Vpr-induced centrosome overduplication (Fig. 4c and Supplementary Figs. 4c and 6e). In addition, TZM-bl, CEM-SS, and primary CD4⁺ T cells infected with HIV-1 WT, but not the Plk4 binding-defective HIV-1 Vpr(1–79) mutant, failed to induce centrosome overduplication. These data suggest that the role of Plk1 in the Vpr-induced centriole duplication is likely minor. Since Plk1 is involved in centriole disengagement, it is possible that the VprBP-Plk1 interaction functions in conjunction with the VprBP-Plk4-dependent centriole duplication. This could be a direction for future research.

REVIEWERS' COMMENTS

Reviewer #1 (Remarks to the Author):

The authors mostly responded to my comments with as much experimentation or reasonable discussion as possible. Thus, I think that the quality of this manuscript has been greatly improved and is now worthy of publication in Nature Communications.

Reviewer #2 (Remarks to the Author):

The authors have added more healthy controls and some longitudinal samples, which were some of our main concerns. The quality of the microscopy has also improved. They opted out of analysing non-cell autonomous Vpr effects, but the editors have, not unreasonably, allowed this.

Reviewer #3 (Remarks to the Author):

The authors addressed appropriately all the points raised in my initial review and underlined the limitations of analyzing centrosome amplification specifically in HIV-infected CD4+ T cells isolated from infected patients. However, their finding that centrosome amplification is decreased in patients upon ART treatment suggests that active HIV replication is an important component driving this process. The addition of experimental evidence in the revised manuscript has much improved the study and strengthen the conclusions of the manuscript.

There are still minor points that the author should addressed:

- 1- Supplementary figure 3b: Flag Plk4 is not detected in the input while it is clearly detected in figure 3c. Could the author comment on this?
- 2- Figure 3 legend: the title should not mention Vpx since the capacity of Vpx to form a ternary complex with VprBP and Plk4 is analyzed in the revised version in supplementary figure 3g.
- 3- Supplementary figure 3g: it is unclear why HIV-2 Vpr NWK08 is tested in the presence of HIV-2 Vpx Rod as shown in the input and the IP GFP? This likely reflects an error in the labelling.

Reviewer #4 (Remarks to the Author):

In this revised manuscript, the authors added new data that addressed a large part of my concerns. The authors should be commended by this hard work. The manuscript has markedly improved. Only a few points should still be addressed:

1. As expected, the authors demonstrated that population of CD4+ T cells with multi-centrosomes is reduced in the ART-treated patients in this revision. In addition to free Vpr protein in blood (page 9), the authors should think and discuss the possibility that active reservoirs may produce viral proteins such as Vpr during ART-treatment (PMID: 37708852; PMID: 32029589; PMID: 37708853).
2. Authors now indicate in this revised paper that MOI 1 is used. Since we know that the majority of cells in patients harbor 1 provirus, relevant infections occur at MOI<1 in vivo. It is also

important to report the phenotype of HIV-1 infection inducing multicentrosomes in T-cells under such conditions, such as MOI 0.3.

REVIEWER COMMENTS

Reviewer #1 (Remarks to the Author):

The authors mostly responded to my comments with as much experimentation or reasonable discussion as possible. Thus, I think that the quality of this manuscript has been greatly improved and is now worthy of publication in Nature Communications.

No additional inquiries.

Reviewer #2 (Remarks to the Author):

The authors have added more healthy controls and some longitudinal samples, which were some of our main concerns. The quality of the microscopy has also improved. They opted out of analysing non-cell autonomous Vpr effects, but the editors have, not unreasonably, allowed this.

No additional inquiries.

Reviewer #3 (Remarks to the Author):

The authors addressed appropriately all the points raised in my initial review and underlined the limitations of analyzing centrosome amplification specifically in HIV-infected CD4+ T cells isolated from infected patients. However, their finding that centrosome amplification is decreased in patients upon ART treatment suggests that active HIV replication is an important component driving this process. The addition of experimental evidence in the revised manuscript has much improved the study and strengthen the conclusions of the manuscript.

There are still minor points that the author should addressed:

1- Supplementary figure 3b: Flag Plk4 is not detected in the input while it is clearly detected in figure 3c. Could the author comment on this?

Response: For Fig. 3c, co-immunoprecipitation (co-IP) was performed by pulling down VprBP (ligand) and examining coprecipitating Plk4 and Vpr. Therefore, the amount of Plk4 coprecipitated is less than 1% of the input. Supplementary Fig. 3b is the result of a reciprocal co-IP analysis performed by pulling down

Plk4 (ligand) and examining coprecipitating VprBP and Vpr. Under these latter conditions, the amount of Plk4 ligand is very high, thus requiring a short exposure. These two reciprocal experiments confirm the association of Vpr, VprBP, and Plk4. To incorporate this reviewer's comment, we added a line in the Supplementary Fig. 3 legend: "Note that Plk4 in the input was not detected because the exposure time required for detecting the Plk4 ligand was short."

2- Figure 3 legend: the title should not mention Vpx since the capacity of Vpx to form a ternary complex with VprBP and Plk4 is analyzed in the revised version in supplementary figure 3g.

Response: As this reviewer commented, we removed "Vpx" from the Fig. 3 legend. This mistake was made when we relocated the current Supplementary Fig. 3f, placed in the main Fig. 3 in the first version.

3- Supplementary figure 3g: it is unclear why HIV-2 Vpr NWK08 is tested in the presence of HIV-2 Vpx Rod as shown in the input and the IP GFP? This likely reflects an error in the labelling.

Response: HIV-2 Vpr NWK08 is correct. It was included in the coimmunoprecipitation analysis shown in Supplementary Fig. 3g, because reviewer #4 wanted to see how HIV-2 Vpr, not HIV-2 Vpx, interacts with VprBP and Plk4. Note that both performed poorly in forming a complex with VprBP and Plk4, further strengthening the specificity of forming the HIV-1 Vpr-VprBP-Plk4 complex. This finding is not surprising considering that HIV-2 Vpr NWK08 and HIV-2 Vpx ROD show only 54% and 49%, respectively, identities with HIV-1 Vpr.

Reviewer #4 (Remarks to the Author):

In this revised manuscript, the authors added new data that addressed a large part of my concerns. The authors should be commended by this hard work. The manuscript has markedly improved. Only a few points should still be addressed:

1. As expected, the authors demonstrated that population of CD4+ T cells with multi-centrosomes is reduced in the ART-treated patients in this revision. In addition to free Vpr protein in blood (page 9), the authors should think and discuss the possibility that active reservoirs may produce viral proteins such as Vpr during ART-treatment (PMID: 37708852; PMID: 32029589; PMID: 37708853).

Response: As suggested, we cited two papers mentioned by this reviewer and added a sentence "These findings are in line with the notion that active reservoirs produce viral proteins, such as Vpr during ART treatment^{78,79}" on page 9.

2. Authors now indicate in this revised paper that MOI 1 is used. Since we know that the majority of cells in patients harbor 1 provirus, relevant infections occur at MOI<1 in vivo. It is also important to report the phenotype of HIV-1 infection inducing multicentrosomes in T-cells under such conditions, such as MOI 0.3.

Response: The editor kindly allowed us to opt out of addressing this question.